

# Intercomparison of mid-latitude tropospheric and lower stratospheric water vapor measurements and comparison to ECMWF humidity data

Stefan Kaufmann[1], Christiane Voigt[1,2], Romy Heller[1], Tina Jurkat-Witschas[1], Martina Krämer[3], Christian Rolf[3], Martin Zöger[4], Andreas Giez[4], Bernhard Buchholz[5], Volker Ebert[5], Troy Thornberry[6,7], Ulrich Schumann[1]

[1]Deutsches Zentrum für Luft- und Raumfahrt, Institut für Physik der Atmosphäre, Oberpfaffenhofen, 82234, Germany
[2]Johannes Gutenberg-Universität, Institut für Physik der Atmosphäre, Mainz, 55128, Germany
[3]Forschungszentrum Jülich, Institute for Energy and Climate Research (IEK-7), Jülich, 52428, Germany
[4]Deutsches Zentrum für Luft- und Raumfahrt, Flight Experiments, Oberpfaffenhofen, 822234, Germany
[5]Physikalisch-Technische Bundesanstalt Braunschweig, Braunschweig, 38116, Germany
[6]NOAA Earth System Research Laboratory, Chemical Sciences Division, Boulder, Colorado, USA
[7]Cooperative Institute for Research in Environmental Sciences, University of Colorado Boulder, Boulder, Colorado, USA

*Correspondence to*: Stefan Kaufmann (Stefan.Kaufmann@dlr.de)

**Abstract.** Accurate measurement of water vapor in the climate sensitive region near the tropopause turned out to be very challenging. Unexplained systematic discrepancies between measurements at low water vapor mixing ratios made by different instruments on airborne platforms have limited our ability to adequately address a number relevant scientific questions on the humidity distribution, cloud formation and climate impact in that region. Therefore, during the past decade, the scientific community has undertaken substantial efforts to understand these discrepancies and improve the quality of water vapor measurements. This study presents a comprehensive intercomparison of airborne state-of-the-art in situ hygrometers deployed onboard the DLR (German Aerospace Center) research aircraft HALO during the Mid-Latitude CIRRUS (ML-CIRRUS) campaign conducted in 2014 over central Europe. The instrument intercomparison shows that the hygrometer measurements agree within their combined accuracy (±10 to 15%, depending on the humidity regime), total mean values even agree within 2.5%. However, systematic differences on the order of 10% and up to a maximum of 15% are found for mixing ratios below 10 parts per million (ppm) $H_2O$. A comparison of relative humidity within cirrus clouds does not indicate a systematic instrument bias in either water vapor or temperature measurements in the upper troposphere. Furthermore, in situ measurements are compared to model data from the European Centre for Medium-Range Weather Forecasts (ECMWF) which are interpolated along the ML-CIRRUS flight tracks. We find a mean agreement within ±10% throughout the troposphere and a significant wet bias in the model on the order of 100% to 150% in the stratosphere close to the tropopause. Consistent with previous studies, this analysis indicates that the model deficit is mainly caused by a blurred humidity gradient at tropopause altitudes.



## 1. Introduction

Water vapor is one of the most important trace gases in Earth's atmosphere due to its large influence on the radiation budget and atmospheric dynamics. It absorbs and emits infrared radiation throughout the entire profile of the atmosphere (Kiehl and Trenberth, 1997). The radiative effect of small changes in water vapor concentration is most pronounced in the upper

troposphere and lower stratosphere (UTLS) where absolute $H_2O$ mixing ratios are two to four orders of magnitude lower than on the ground (e.g., Ramanathan and Inamdar, 2006; Solomon et al., 2010; Riese et al., 2012). Besides the direct radiative effect, water vapor also provides one of the strongest feedback parameters to temperature changes in the atmosphere (Manabe and Wetherald, 1967; Dessler et al., 2008).

Additionally, water vapor is the most important parameter for cloud formation and their lifetime. From an energy

perspective, clouds not only influence the radiation balance but also redistribute energy through latent heat during condensation and evaporation. In a changing climate, changes in latent heat fluxes influence global dynamics like, e.g., the Hadley circulation and extratropical storm tracks (Schneider et al., 2010). The radiative effect of clouds is much more complex than the effect of greenhouse gases due to very inhomogeneous cloud cover as well as microphysical and radiative properties of clouds at different altitudes. The countervailing effects of the reflection of solar shortwave radiation and the

trapping of longwave radiation determines the net radiative effect of clouds, whether cooling or heating, depending on cloud properties, surface albedo, sun elevation, etc. (e.g., Liou, 1986; Lynch, 1996; Lee et al., 2009).

The various atmospheric processes related to water vapor impose challenges for its measurement. Radiative effects are directly linked to the concentration of (gaseous) molecules, e.g., in the lower stratosphere, where clouds rarely occur. Regarding clouds, the main control parameter is the relative humidity with respect to liquid water or ice (RHw and RHi,

respectively). In consequence, the measurement accuracy and resolution required to improve our understanding of the atmosphere strongly depends on the research question. Regarding the radiative effect of stratospheric $H_2O$, the main challenge is the absolute accuracy at mixing ratios below 10 parts per million (ppm, equivalent to µmol/mol) since small changes of less than 1 ppm significantly impact the radiation budget (Solomon et al., 2010). For cloud effects, the challenge is even bigger, especially in very cold ice clouds where ice supersaturation and cloud properties are strongly linked (Jensen

et al., 2005; Shilling et al., 2006; Krämer et al., 2009). A 10% difference in RHi, which falls within the combined uncertainty in water vapor and temperature measurements, can result in substantially different cloud properties.

During the past several decades, a number of $H_2O$ measurement intercomparisons during field campaings including aircraft in situ, balloon-borne and satellite instruments revealed that the measurement uncertainty was significantly higher than 10%, even occasionally exceeding 100% in at the lowest mixing ratios of the UTLS (e.g. Oltmans et al., 2000; Vömel et al., 2007;

Weinstock et al., 2009). These large discrepancies motivated the comprehensive intercomparison campaign AquaVIT-1 at the AIDA (Aerosol Interaction and Dynamics in the Atmosphere) cloud chamber in Karlsruhe 2007 (Fahey et al., 2014) and the follow-up campaigns AquaVIT-2 and -3 in 2013 and 2015, respectively. In the controlled environment of the cloud chamber, the agreement between the instruments was improved compared to the measurements on the different airborne



platforms but still in the 20% range for mixing ratios between 1 and 10 ppm. As a consequence, novel concepts and instruments (e.g. Thornberry et al., 2013; Kaufmann et al., 2014; Kaufmann et al., 2016; Buchholz et al., 2017) and improved techniques for inflight (Rollins et al., 2011) and ground calibration (Meyer et al., 2015) were developed to improve the accuracy of $H_2O$ measurements.

Since space and measurement time on research aircraft are limited and expensive, intercomparable airborne data sets of water vapor measurements are scarce. The most recent comprehensive intercomparison was conducted in 2011 on the NASA WB-57 high altitude aircraft during the MACPEX campaign (Rollins et al., 2014). Similar to the present study, five different hygrometers using differing water vapor detection techniques were mounted on the aircraft. In the dry regime below 10 ppm, instruments were found to typically agree within their stated combined accuracies. However, the authors argue that the

remaining discrepancies are very likely of systematic nature and result from undetermined offsets in flight (Rollins et al., 2014). Referring to the accuracy required to address the questions noted above, it seems that while significant progress has been made in recent years, there is still some way ahead towards answers to scientific questions not being limited by measurement accuracy.

The aim of this study is to provide another step towards a better understanding of the accuracy of airborne water vapor

measurement. We present a comprehensive intercomparison of the major state of the art hygrometers operated by the German research community. This unique data set is used to assess the performance of the individual instruments and to provide a solid base for comparison to the Integrated Forecast System (IFS) of the European Centre for Medium-Range Weather Forecasts (ECMWF). Section 2 briefly describes the ML-CIRRUS campaign during which five independent in situ hygrometers were operated simultaneously. Section 3 provides a summary of the different instruments. The methodology of

the intercomparison is described in section 4 while the intercomparison itself is discussed in section 5. In addition, this section also includes a comparison of relative humidity inside of cirrus clouds as well as an intercomparison of in situ measurements with ECMFW IFS model data.

## 2. ML-CIRRUS campaign

The ML-CIRRUS campaign with the DLR research aircraft HALO took place in March and April 2014 with the aircraft

based in Oberpfaffenhofen, Germany. A detailed summary of the scientific goals, the flight strategy and the instrumentation is given in Voigt et al. (2017). During the campaign period, HALO performed 16 research flights with 88 flight hours in total. The flights were designed for a comprehensive characterization of mid-latitude cirrus and contrail cirrus using in situ as well as remote sensing instruments. ML-CIRRUS aimed for a better understanding of cirrus cloud formation in different meteorological conditions (Krämer et al., 2016; Luebke et al., 2016; Wernli et al., 2016; Urbanek et al., 2017) to improve

our estimation of the radiative impact of cirrus (Krisna et al., 2018) as well as for air traffic impacts on high cloud cover (Schumann et al., 2017; Grewe et al., 2017). Therefore, the flight plans were mainly designed to obtain a maximum number of flight hours either within cirrus clouds for in situ measurements or approximately 1 km above cirrus for lidar and



dropsonde measurements. The implications of the flight strategy on the water vapor intercomparison are discussed in section 4.1. Looking for cirrus cloud life cycle under different meteorological conditions, the flights covered almost the entire region of central Europe from the northern British coast down to Portugal (Figure 1).

To achieve the scientific goals of the mission, the HALO payload for ML-CIRRUS instruments to measure cloud particles,
aerosols, trace gases and dynamic parameters. The aircraft cabin was equipped with several novel in situ instruments for trace gases and aerosols, dropsondes and a Differential Absorption Lidar (DIAL) system for water vapor and cloud measurements. Furthermore, cloud particles and aerosols were measured in situ using a set of nine wing probes. Since this paper focusses on the intercomparison of the in situ water vapor measurements during ML-CIRRUS, only those instruments will be described here in detail. A full list of instruments, their descriptions and references can be found in Voigt et al.
10   (2017).

## 3   Instruments

The HALO payload for ML-CIRRUS included five different water vapor instruments, which provides the opportunity to compare different measurement methods and a comparison of both gas phase and total water measurements. In particular, three completely independent measurement principles for water vapor were utilized: mass spectrometry (AIMS-$H_2O$),
Lyman-$\alpha$ photofragment fluorescence spectroscopy (FISH) and tunable diode laser absorption spectroscopy (SHARC, HAI and WARAN). While AIMS-$H_2O$ and SHARC measured gas phase water vapor via a backward facing inlet, FISH, HAI and WARAN measured total water (gas phase + evaporated cloud particles) using forward facing inlets. A summary of key parameters for each instrument is given in Table 1.

### 3.1   AIMS-$H_2O$

The Atmospheric Ionization Mass Spectrometer for water vapor (AIMS) is a linear quadrupole mass spectrometer designed to measure low water vapor mixing ratios typical for the upper troposphere and lower stratosphere (Kaufmann et al., 2016; Thornberry et al., 2013), and, in a different configuration, HCl, $HNO_3$ and $SO_2$ (Jurkat et al., 2016). The instrument samples gas phase water vapor through a backward facing heated inlet. After passing a pressure regulation valve, sample air is directly ionized in an electrical discharge ion source. Inside the ion source multiple ion-molecule reactions form
$H_3O^+(H_2O)_n$ ion clusters with n = 0…2. The abundance of these ion clusters is then measured by the mass spectrometer and used to quantify the original water vapor molar mixing ratio in the ambient air. In order to accurately link the ion count rate with the $H_2O$ mixing ratio, the instrument is calibrated in flight by regularly adding a water vapor standard generated by the catalytic reaction of hydrogen and oxygen to form $H_2O$ on a heated Pt surface (Rollins et al., 2011). AIMS operates at a measurement range between 1 and 500 ppm with an overall accuracy of 7-15%, mainly depending on the actual water vapor
concentration (Kaufmann et al., 2016). During ML-CIRRUS ambient air was sampled through 8.5 mm ID Synflex tubing and a bypass flow was used to reduce the residence time of air in the inlet line to below 0.2 s. This results in a real





measurement frequency of ~ 4 Hz corresponding to around 50 m horizontal resolution. In order to achieve the best possible accuracy of the instrument it was calibrated once or twice during each research flight. The stability of the calibration standard was guaranteed by six ground reference measurements against a MBW 373LX dew point mirror during the campaign period.

## 3.2 FISH

FISH (Fast In situ Stratospheric Hygrometer) is a closed cell Lyman-α photofragment fluorescence hygrometer which has been operated on various research aircraft for more than 20 years (Meyer et al., 2015; Schiller et al., 2009). The operating principle of the instrument is described in detail by Zöger et al. (1999). It uses the Lyman-α radiation of an UV lamp at 121.6 nm to dissociate water molecules into single H atoms and an excited-state OH molecules. Returning to the ground state, the OH molecules emit radiation at a wavelength between 285 and 330 nm. The intensity of this radiation is proportional to the water vapor molar mixing ratio in the measurement cell and is quantified using a photomultiplier tube. FISH is calibrated regularly on the ground to relate the measured signal to the water vapor mixing ratio using a DP30 dew point mirror as reference instrument. A detailed description of the calibration procedure can be found in Meyer et al. (2015). FISH is able to measure water vapor mixing ratios in a range from 1 to 1000 ppm. The overall uncertainty during ML-CIRRUS was determined to be 6% relative and ±0.4 ppm absolute offset uncertainty. FISH was connected to a forward-facing inlet to sample total water. The pressure difference between inlet (static + dynamic) and gas exhaust (only static) ensures a flow rate >10 standard L min$^{-1}$ and thus allows for fast measurements in UTLS and cirrus conditions.

## 3.3 SHARC

SHARC (Sophisticated Hygrometer for Atmospheric ResearCh) is a tunable diode laser (TDL) hygrometer developed at DLR Flight Experiments. It is a closed cell hygrometer which uses the absorption line of water vapor at 1.37 µm. To cover a wide humidity range SHARC uses a dual path Herriott type cell with a single pass absorption length of approximately 0.17 m and a multi pass absorption length of approximately 8 m. The cell is completely fibre coupled to minimize parasitic absorption outside the measurement volume and has a very compact volume of 83 cm$^3$. The measurement range is from 10 to 50000 ppm, constrained by the detection limit of the absorption signal at low water vapor mixing ratios. The overall uncertainty is 5% relative and ±1 ppm absolute offset uncertainty. SHARC was operated with a 6.35 mm backward facing stainless steel inlet during ML-CIRRUS sampling gas phase $H_2O$ with a total flow of 15 standard L min$^{-1}$ at ground decreasing to 1.5 standard L min$^{-1}$ at highest flight levels. The real time data reduction uses a multi-line Voigt fit at 5 Hz to calculate the water vapor mixing ratio. For the intercomparison, the data were averaged to 1 Hz.



### 3.4    HAI

HAI (Hygrometer for Atmospheric Investigations) is a four channel TDL hygrometer which uses two different absorption wavelengths (1.37 µm and 2.6 µm) in both closed and open cell geometries (Buchholz et al., 2017). HAI uses a complete physical model in combination with spectral water absorption line parameters mostly measured at the Physikalisch

Technische Bundesanstalt Braunschweig (PTB) (Pogány et al., 2015) and monitors pressure, temperature and absorption path length in order to calculate the water vapor concentration for a given absorption spectrum without prior calibration. The accuracy of this approach was verified recently by a side by side comparison (Buchholz et al., 2014) of a previous PTB laser absorption spectrometer with the German national primary humidity standard. HAI has 1.5 m optical path length for the closed cell and 4.2 m for the open path. For this work, we use data from the 1.37 µm closed cell channel of HAI in the range

of 20 to 40000 ppm. The overall uncertainty for this channel is 4.3% relative and ±3 ppm absolute offset uncertainty. The closed cell was connected to a 12.7 mm forward facing stainless steel inlet and was actively pumped. The effective time resolution of the instrument is 0.7 s corresponding to a spatial resolution at flight altitude of around 150 m.

### 3.5    WARAN

The WARAN (WAter vapoR ANalyzer) instrument consists of a commercial WVSS-II (SpectraSensors Inc., USA) tunable

diode laser instrument in combination with a custom inlet and an additional pump for the flow through the measurement cell (Kaufmann et al., 2014; Groß et al., 2014). While the instrument was operated on other campaigns parallel to a frostpoint hygrometer (Heller et al., 2017), during ML-CIRRUS the WARAN was integrated in the AIMS rack and connected to a forward-facing inlet to sample total water. The inlet pylon was the same as used for AIMS-H$_2$O. As for the other instruments operating with a forward facing inlet, only cloud free measurement sequences are used for the intercomparison. Due to the

high detection limit of the instrument ($> 50$ ppm, stated by the manufacturer), the intercomparison of this instrument is limited to tropospheric conditions. During ML-CIRRUS the WARAN was mainly used to detect cloud water. Due to the enhancement of ice particles in the inlet by a factor between 20 and 35, measured total water mixing ratios are relatively high (Afchine et al., 2017). Hence the instrument detection limit allows for cloud water quantification for most clouds except for very thin cirrus.

### 25   3.6    Additional instrumentation

For data evaluation with respect to relative humidity, cloud detection and model intercomparison, we use a couple of other parameters measured on board of HALO during ML-CIRRUS. Static pressure and static temperature are measured by the Basis HALO Measurement and Sensor System (BAHAMAS, Krautstrunk and Giez, 2012; Giez et al., 2017). The accuracy of the pressure sensor is 0.3 hPa, accuracy of the static temperature measurement is 0.5 K. The SHARC hygrometer (see

section 3.3) is also part of BAHAMAS. Cloud detection was done using data of the Cloud and Aerosol Spectrometer with Detection of Polarization (CAS-DPOL) which was mounted under the wing of HALO (Baumgardner et al., 2001; Voigt et





al., 2017). The cloud probe measures particles in a size range between 0.5 µm and 50 µm and is thus sensitive to natural cirrus as well as contrail ice particles.

## 4 Methodology and conditions for intercomparison

The intercomparison of the different water vapor instruments can be approached in various ways. Starting from a comparison of the directly measured time series and profiles (Figure 2) we further use the dataset from the entire ML-CIRRUS campaign, similar to previous approaches, e.g., by Rollins et al. (2014), Fahey et al. (2014) or Meyer et al. (2015). This section describes the framework of the intercomparison and the methodology of the data evaluation including the determination of a water vapor reference value.

### 4.1 Flight strategy

A discussion of the flight strategy during ML-CIRRUS is important since the campaign did not aim for a statistically uniform sampling in terms of water vapor but on the investigation of cirrus clouds. The flight patterns typically consists of three components: (1) sampling inside cirrus clouds in order to obtain in situ information on particle distribution and their interaction with trace gases and aerosols, (2) remote sensing segments of cirrus clouds by lidar and radiation measurements where HALO flew ~1 km above the cirrus and (3) transfer flight segments to approach specific weather systems like warm conveyor belts or mountain lee wave regions over western Europe (dark blue and magenta flight tracks in Figure 1). In total, we have around 160000 1 Hz data points in the UTLS with $H_2O$ mixing ratios between 3 and 1000 ppm. Of those data points, approximately 22% are in stratospheric conditions ($\theta > 350$ K), and 33% are in-cloud measurements.

The dedicated search for cirrus conditions leads to a higher detection frequency of both, in-cirrus and above cirrus sampling relative to their natural occurrence. Since inside cirrus the relative humidity is expected to be distributed around saturation, this allows for an independent check of the absolute values of the gas phase water vapor measurements. However, extensive in situ sampling in cirrus limits the data for intercomparison of total and gas phase instruments. The remote sensing legs and the transfer segments provide a comprehensive water vapor dataset within the lower stratosphere. The lidar requires a certain vertical distance to the cirrus upper edge, hence most of the stratospheric data were sampled roughly 1 km above that level. Directly above cirrus level less data points are sampled. During the transfer segments, flight altitude and horizontal position of the aircraft are independent of meteorological conditions; however, due to the typical high flight altitude of HALO, most of these data points are within the lower stratosphere.

Overall, the ML-CIRRUS flight strategy shifts the sampling of water vapor compared to un-biased sampling of the UTLS in a way that there is a higher detection frequency of humid upper tropospheric air within cirrus clouds, higher detection frequency of stratospheric measurements at a distance of around 1 to 1.5 km to the tropopause and only a small detection frequency of data in dry tropospheric conditions and directly above the tropopause. However, the measurement strategy



should only affect the amount of data in certain water vapor ranges and not the performance of each instrument within its specification..

## 4.2 Data processing and filtering

In order to construct a consistent data set from all five water vapor instruments on board HALO, the specific time resolutions
and response characteristics are considered for each instrument. The goal is to retain as much information as possible while minimizing data processing related artefacts. Since all instruments reported data either with a non-uniform frequency or on 1 Hz intervals, the latter was used to unify the data. For AIMS, the 1 Hz data are created by averaging over three data points. Data from FISH are on a 1 Hz integer time base. For SHARC and HAI, the 1 Hz resolution data are interpolated onto integer values. The only instrument with a lower time resolution than 1 Hz is the WARAN with ~0.4 Hz. Since it is not useful to
interpolate this data set onto a 1 Hz interval, each measured value is assigned to the closest integer time value. This processing allows comparison of the $H_2O$ measurements directly without imposing any substantial interpolation artefacts in the measured values which could affect the interpretation of the intercomparison.

Since three instruments (FISH, HAI and WARAN) measured total water, cloud sequences were filtered out for the comparison of gas phase $H_2O$. The cloud filtering was done in a two-step process using both the total water measurements
themselves and cloud probe particle measurements by the CAS-DPOL. To make sure that in-cloud data are definitely filtered out, all data with total water concentrations above saturation are flagged as "in-cloud". However, this implies that supersaturated cloud-free conditions are left out as well. As quality check for the filtering procedure, particle concentrations measured by the CAS-DPOL are used to double check the cloud mask. In this step, very few additional data points are rejected which might be due to very thin sublimating clouds or the different positions of cloud probe under the wing and
water vapor inlets at the top fuselage.

Further data filtering was applied manually in order to clear data that suffer from obvious sampling artefacts. Concerning AIMS, the pressure regulation of the instrument (Kaufmann et al., 2016) during ML-CIRRUS was not fast enough to compensate for the pressure drop during the fast first ascent on each flight. For this reason, $H_2O$ data in that region are not reliable and left out. Furthermore, there are a few ascent and descent sequences where one or more instruments showed a
significant time lag of a couple of seconds compared to the other instruments. The causes of these lags and their intermittent occurrence are not clear and the respective sequences are filtered out.

## 4.3 Reference value

The determination of a reference value for the intercomparison is guided by various considerations. One possibility is the agreement on a common reference instrument. The airborne intercomparison during MACPEX (Rollins et al., 2014), e.g.,
used the Harvard Lyman-α as single instrument reference. However this approach is complicated for the instrument combination deployed during ML-CIRRUS since there was no instrument on HALO which measured gas phase $H_2O$ and



simultaneously covered the complete range of mixing ratios. For that reason, we follow the approach of the AQUA-VIT campaign described in Fahey et al. (2014) where the mean value of a set of core instruments was used as reference. This allows for a combined intercomparison of data in the lower stratosphere (AIMS, FISH) and in the upper troposphere in cirrus clouds (AIMS, SHARC) and clear sky (AIMS, FISH, SHARC, HAI). We further compare the middle troposphere at higher

$H_2O$ mixing ratios (SHARC, HAI, WARAN). The reference value for each 1 s step is calculated as the mean of AIMS, FISH, SHARC and HAI data points with the condition that at least two instruments provided valid data for a single point. Data from the WARAN are not included in the reference calculation since their uncertainty is significantly higher.

## 5    Intercomparison

The basis for the intercomparison of $H_2O$ data during ML-CIRRUS are time series from each instrument, an example sequence of which is shown in Figure 2 (c) for the flight on 3 April 2014. For all total water instruments, only cloud-free data are used for the intercomparison. This flight aimed for in situ and remote measurements of thin cirrus over Germany which were potentially influenced by Saharan dust (Weger et al., 2018). Flight altitude and water vapor mixing ratios in Figure 2 show the alternation of tropospheric in situ legs ($H_2O$ ~30…120 ppm) and LIDAR legs in the stratosphere ($H_2O$

~5 ppm). Except for the WARAN which seems to measure too high at the beginning of the flight, all instruments agree reasonably well in both upper troposphere and lower stratosphere. Figure 2 (a) shows a profile for the upper troposphere and lower stratosphere using data from the second descent (indicated by dotted lines in Figure 2 (c)). The instruments follow the same structures in both regions with a much higher variation in $H_2O$ mixing ratios in the upper troposphere. The agreement also holds for the second profile down to 3 km altitude (Figure 2 (b)) however mixing ratios there are too high to be

measured by AIMS and FISH. The short ascent to 8 km after the profile shows a significant deviation between SHARC, HAI and WARAN. Both total water instruments (HAI and WARAN) measure higher values than the SHARC which is most likely due to wet contamination of their measurement cells when encountering liquid clouds during the descent. Sequences with such contamination are identified for the entire data set and filtered out for the intercomparison.

### 5.1    Correlation of single instruments

To investigate the overall performance of the different measurement systems, twelve ML-CIRRUS flights were combined similar to the one shown in Figure 2. This complete data set is used to produce the scatter plots in Figure 3, where selections of four combinations of instrument pairs are displayed. The scatter plot of AIMS and FISH in Figure 3 (a) shows a very close correlation from below 4 ppm up to ~600 ppm corresponding to the upper limit of AIMS. For stratospheric mixing ratios below 10 ppm the correlation broadens with AIMS exhibiting a tendency to higher humidity values and FISH to lower

humidity values. Figure 3 (b) shows the correlation between AIMS and SHARC, the two instruments measuring solely gas phase $H_2O$ and thus the only correlation plot where in-cloud data are displayed together with clear sky data. Consistent with the Figure 3 (a) this correlation is very narrow, slightly widening only for the high concentrations at the upper AIMS





measurement limit. A similar narrow correlation is found for HAI versus FISH (Figure 3 (c)) from 20 ppm up to 1000 ppm. For all three scatterplots (Figure 3 (a)–(c)) correlation coefficients are higher than 0.99. In contrast to panels (a)–(c), Figure 3 (d) spans the range to higher humidity from 10 to 10000 ppm displaying data from WARAN and SHARC. Between 100 and 300 ppm, the WARAN shows a slight dry bias which disappears for higher mixing ratios. Compared to the other

instruments, the WARAN exhibits a significantly larger scatter with complete sequences lying way above the one-to-one line. These sequences are associated with initial ascent during the flights, where the WARAN occasionally shows a wet bias (data points marked orange in Figure 3 (d)). These data points are omitted from the intercomparison. The dry bias and larger scatter are also reflected in the correlation coefficient which is 0.94 for Figure 3 (d). The comparison with WARAN measurements during other campaigns suggests that the deviations are likely caused by systematic offsets in the original

calibration of the instrument. Thus, the analysis is probably only valid for this specific instrument during the ML-CIRRUS campaign. Overall, the correlation plots indicate a good agreement for AIMS, FISH, SHARC and HAI throughout the entire campaign.

## 5.2    Deviation with respect to reference value

In order to quantify the performance of each instrument, the deviations of each instrument from the reference value (see

section 4.3) are displayed in Figure 4, similar to previous studies (Fahey et al., 2014; Rollins et al., 2014). On the x-axis, the $H_2O$ reference value is shown. The y-axis denotes the relative difference for each instrument from that reference value. The small dots are the measured 1 Hz values, the big symbols are mean values for logarithmic bins in $H_2O$. Additionally, the broad bars represent the interquartile range in each bin and the narrow bars are the 10/90 percentiles. In the grey box on the left, mean values and respective percentiles for the entire data set of each instrument are shown. As shown in Table 2, the

mean deviations of AIMS, FISH, SHARC and HAI are below 2.5%, indicating that there is no consistent systematic bias in any single instrument. The situation looks different for the WARAN instrument where the dry bias at low $H_2O$ mixing ratios can be clearly seen in the $H_2O$ resolved deviation but not in the overall mean (Figure 4 (e)).

Looking at Figure 4 (a) and (b) in more detail, the agreement between AIMS and FISH in the lower stratosphere below 10 ppm seems good with single values of both instruments mostly falling within ±15%. Since these are the only two instruments

measuring in the low ppm range, the plot is a direct comparison of both instruments. In fact, there is a systematic difference between both instruments between 4 and 10 ppm. In that region the mean values of the instruments differ by 4 to 16% with AIMS measuring higher and FISH measuring lower mixing ratios. Interestingly, the difference between the instruments for the driest conditions (3.5 to 4.5 ppm) is smaller than for the next several bins (2.4% versus 6.5%). Examining all of the time series plots from the campaign (not shown), there are some distinct stratospheric legs where AIMS is up to 1 ppm higher

than FISH (corresponding to a relative deviation of ~20%). The reason for this deviation is not completely clear; one explanation could be a contamination of the AIMS vacuum system. However, it is unlikely that this is the only cause since the behavior changes occasionally from one leg to another within the same flight. For upper tropospheric measurements



(where more than the two instruments contribute to the reference value), the agreement of the mean values with the reference is better than 5%. The same holds for the SHARC measurements (Figure 4 (c)) throughout its complete range with a slight tendency to lower mixing ratios (3-4%) compared to the reference between 30 and 200 ppm. HAI data (Figure 4 (d)) also fall in the same range of variation with mean values being consistently slightly higher by about 3% than the reference value in

the range between 30 and 2000 ppm. For both SHARC and HAI, the single measurement scatter is within ±20% with respect to the reference. Considering the fact that all four instruments contribute to the reference value, one can state that FISH and SHARC tend to consistently report slightly lower mixing ratios than AIMS and HAI. The WARAN measurements (Figure 4 (e)) fall off compared to the other four instruments, exhibiting a significant low bias for mixing ratios below 300 ppm. However, these data are still within the uncertainty specifications of the instrument (see Table 1).

## 5.3    Comparison of relative humidity in clouds

The comparison of relative humidity measurements in clouds can be considered as a further measure for the quality of the $H_2O$ measurements which is independent from any kind of reference value. In contrast to measurements in liquid clouds, much stronger deviations of RHi from saturation are possible in ice clouds due to their higher thermodynamic inertia. Relative humidity with respect to ice (RHi) inside cirrus clouds can be very variable due to advection as well as small scale

turbulence inside the cloud (e.g. Gettelman et al., 2006; Petzold et al., 2017). However, if the measurements include a sufficiently even sampling of meteorological conditions, a distribution of RHi with mode value close to 100% would be expected. In order to calculate RHi from the measured $H_2O$ mixing ratios, we have used the static temperature and static pressure measurements onboard HALO to calculate water vapor partial pressure and saturation pressure. The saturation pressure over ice is calculated using equation (7) from Murphy and Koop (2005).

Here, we compare in-cloud measurements of RHi for the two water vapor instruments with backward facing inlets, AIMS and SHARC (see also Figure 3 (b)). In total, more than 50000 in-cloud data points were acquired during ML-CIRRUS with numbers varying between 2000 and 11000 for individual flights. The frequency distribution of RHi for the entire dataset of the ML-CIRRUS campaign is shown in Figure 5. Data from both instruments are almost normally distributed, with mean values slightly below ice saturation. Fitting a normal distribution to both datasets, they peak at RHi = 97% for AIMS (52700

data points) and 94% for SHARC (56300 data points). The FWHM of the distribution is 26.7% for AIMS and 19.4% for SHARC. Both distributions are slightly asymmetric with a tail towards higher supersaturation which is more pronounced in the SHARC measurements. This agrees with results from Ovarlez et al. (2002), who find similar asymmetric distributions for temperatures below -40°C.

Considering the instrumental uncertainties, both distributions appear reasonable. However, the question remains whether the

slight shift of the RHi distribution relative to ice saturation is caused by systematic instrument biases ($H_2O$ and temperature), inlet issues (e.g., sucking in and evaporating ice particles) or a sampling bias in the flight strategy. If the sampling were biased toward either forming/growing cirrus or evaporating cirrus, one would expect a positive or negative RHi bias with





respect to saturation, respectively. During ML-CIRRUS, individual flights typically targeted specific meteorological conditions, e.g., the updraft region of warm conveyor belts or mountain wave cirrus. Hence, a sampling bias for individual flights is very likely. In order to investigate that, Figure 6 shows the mean values for the in-cloud RHi distributions of AIMS and SHARC including interquartile ranges and 10/90 percentile ranges for each flight. For flights number one to five, AIMS and SHARC deviate by 4-8% with one exception on flight three where the deviation is around 20%. These data originate from a two-step profile through cirrus clouds with high updraft velocities over the Balearic islands. During that flight, there is a systematic difference between AIMS and SHARC which is most pronounced during the two cirrus transects (difference of around 20% compared to 7…10% during the rest of the flight). From the high updraft velocity, one would rather expect supersaturation inside the cirrus. For flight number seven, there is not enough in-cloud data from AIMS to produce a reasonable RHi distribution. For flights number eight, nine and ten, the agreement of both instruments is almost perfect while for the last two flights AIMS tended to measure slightly lower RHi values than SHARC but with a difference of less than 3%. The spread of the RHi measurements is similar for both instruments (AIMS interquartile range 10-20%, SHARC interquartile range 8-17%) with the lower values for SHARC arising from a slightly better precision.

The observed trend could be an indication of instrumental drift over the campaign period, however we cannot state which instrument is subject to a drift. Flights with mean super- or subsaturation are almost evenly distributed for AIMS, while SHARC measurements are slightly sub-saturated, especially during the first half of the campaign. From the present data, we do not have clear evidence for an overall sampling bias during the campaign. A possible bias affecting RHi derived from both instruments could be a bias in the static temperature measurement onboard HALO since we use the same temperature information for both instruments. However, the median and mean values of the distributions deviate by less than 6% from saturation for most of the flights, indicating that temperature is not significantly off.

### 5.4 Comparison to the numerical weather prediction model ECMWF

The extensive ML-CIRRUS in situ dataset of upper tropospheric and lower stratospheric humidity further enables an evaluation of the accuracy of UTLS humidity in the ECMWF (European Centre for Medium-Range Weather Forecasts) numerical weather prediction model. A correct representation of water vapor is crucial for weather and climate prediction via various pathways. Besides the troposphere where water vapor is obviously important for cloud formation and precipitation, also the stratospheric mean state influences the predictability in the troposphere (Douville, 2009). Moreover, biases in modelled stratospheric water vapor can induce a frequently observed cold bias in the extratropics (e.g. Boer et al., 1992; Stenke et al., 2008; Chen and Rasch, 2012).

The model data used for analysis of ML-CIRRUS are provided by the Integrated Forecasting System (IFS) of the ECMWF (IFS Version 40r1). For analysis, we use a combination of analysis data with hourly forecasts starting every 12 h from the analysis at 0 and 12 UTC. The data set covers the region of 60°W to 20°E, and 20°N to 70°N. The model includes 137 vertical model levels, with pressure intervals of 18 hPa near 7 km altitude and 7 hPa near 15 km height. For typical flight



altitudes near 11.5 km (200 hPa) the vertical resolution is around 300 m (10 hPa). The horizontal resolution of the data used is 0.5 degree. Higher horizontal resolution would be available from IFS but would not provide more information due to the hourly time resolution. The data are interpolated linearly to the measurement position for a given HALO position (latitude & longitude) above the WGS84 reference ellipsoid. Vertical interpolation is performed in the logarithm of pressure fields

(which varies more smoothly than pressure) based on the static pressure measured by HALO-BAHAMAS (Schumann et al., 2015). The output frequency is 0.1 Hz along the flight track resulting in a distance of roughly 2 km between adjacent data points. The reference $H_2O$ mixing ratio is averaged accordingly over 10 s intervals. Except for the time resolution, the methodology of the intercomparison of model data and measurements is the same as used in section 5.2, simply treating the interpolated model data as "new" instrument. In Figure 7, the relative deviation of the EMCWF data is plotted against the

measured reference $H_2O$ value (same method as used for Figure 4). The small dots represent the interpolated model data point for each valid reference value (see section 4.3). Similarly to in Figure 4, the black triangles denote bin-wise mean values of the relative difference, the grey bars and whiskers represent the interquartile range and the 10/90 percentile range, respectively. In order to get an idea if the sampled air mass is of stratospheric or tropospheric origin, the individual data points are color coded with potential temperature interpolated from the HALO onboard measurements.

As can be seen in Figure 7, the comparison between the model and measurements is different in two distinct humidity regimes. At the higher tropospheric mixing ratios above 30 ppm, there is a reasonable agreement between mean bin values, and the interquartile range is mostly within ±10%. Despite the remarkably good agreement for the mean values, single values scatter significantly resulting in the 10/90 percentiles of around -30% and +20%, respectively. This and the distribution of mean relative differences suggest a slight bias in that region, with ECMWF being slightly lower. With a mean value near

3%, this bias is very small when considering to the overall scatter of the data and the interpolation of the model onto the flightpath. The interpolation procedure is also the reason for the single data points resembling the shape of a mirrored S. This behavior results from comparing the measurement signal with high spatial variability with the rather smooth model data. When using a logarithmic y-scale and the more variable measured mixing ratio as reference on the x-axis, it results in an S-like shape in the individual data points.

The character of the intercomparison differs for lower mixing ratios below 30 ppm found in the tropopause region and the lower stratosphere. In that region, the model significantly overestimates the humidity. The biggest differences between measurement and model occur at mixing ratios between 5 and 8 ppm, typical values for the region directly above the tropopause. The maximum difference is found in the bin between 5.5 and 6.5 ppm where the mean difference is 115% (statistics from 382 data points). The difference decreases again for mixing ratios below 5 ppm indicating a better agreement

between measurement and model with increasing distance to the tropopause. The mean difference for the driest bin (3.5 to 4.5 ppm with 2383 data points) of 46% is less than half of the more humid neighboring bins. However, it still is significant and positive, meaning that ECMWF shows a systematic wet bias for the entire probed region in the lower stratosphere in spring.



The maximum differences close to tropopause mixing ratios indicate that the difference between measurement and model is rather caused by a blurred humidity gradient in the model tropopause region than by a systematic bias in the (deep) stratospheric humidity of ECMWF. This is in agreement with previous studies, e.g., by Kunz et al. (2014) and Dyroff et al. (2015). The latter study shows a good agreement between measurement and model for vertical distances to the tropopause of
6 km and higher and model wet bias between 2 and 6 km above the tropopause for the extratropics. During ML-CIRRUS, the maximum distance above the tropopause was 3.5 km, hence the measurements are probably not stratospheric enough to leave the wet bias region. However, the trend towards better agreement deeper in the stratosphere can be seen in the color coding in Figure 7 as well as in Table 3 where mean difference are binned by potential temperature rather than the mixing ratio. It turns out, that the wet bias strongly peaks at potential temperatures between 350 K and 360 K (mean difference of
88%) whereas it decreases from there with increasing altitude in the stratosphere (higher potential temperature) as well as into the troposphere (lower potential temperature). One reason for such a deviation could be an incorrect representation of the tropopause height in the model. Comparing flights in rather simple meteorological conditions (and thus a good representation of the tropopause) with more complex situations or situations with influence from Saharan dust, we do not observe any change in the pattern of the humidity intercomparison. Hence, we attribute the observed differences to the
humidity gradient at the tropopause which is sharper than represented in ECMWF, at least for the observed European spring conditions.

## 6    Discussion and summary

We intercompare water vapor measurements from different state of the art in situ instruments onboard the DLR research aircraft HALO during the mid-latitude UTLS field project ML-CIRRUS. It is the first comprehensive intercomparison of all major hygrometers operated by the German research community including three TDL instruments (HAI, SHARC and WARAN), one mass spectrometer (AIMS) and the established Lyman-$\alpha$ hygrometer FISH. The intercomparison includes a large span of humidity conditions from lower stratospheric to lower tropospheric $H_2O$ molar mixing ratios, with different
instruments covering different parts of the mixing ratio spectrum. This work focusses on the intercomparison of gas phase water vapor measurements, meaning that only clear sky data are used from instruments measuring total water (HAI, FISH and WARAN). The flight strategy of ML-CIRRUS focused on the investigation of mid-latitude cirrus cloud with in situ and remote sensing (LIDAR) instrumentation. Hence, the majority of data points originate from the mid-latitude upper troposphere and lower stratosphere above Europe and the western Atlantic in spring 2014.
The agreement between the in situ instruments, expressed by the relative difference to a reference value (mean value of at least two instruments), is generally good and consistent with previous intercomparison studies (Rollins et al., 2014). For all instruments except the WARAN, the overall mean deviation from the reference value is below 2.5%. This is an indication for the successful efforts to improve the accuracy of UTLS $H_2O$ measurements during the past decade, motivated by large



discrepancies that have been found before (Fahey et al., 2014). Still, systematic discrepancies remain between the instruments in specific regimes which need to be addressed in order to improve our understanding of the humidity budget in the lowermost stratosphere or of cirrus formation under very cold conditions (Gao et al., 2004; Krämer et al., 2009; Jensen et al., 2017). One major issue is the difference between FISH and AIMS for stratospheric mixing ratios below 10 ppm. The

observation that the mass spectrometer AIMS measures systematically higher mixing ratios than FISH is similar to the findings during the MACPEX intercomparison (Rollins et al., 2014). During that campaign, the maximum difference of bin mean values is 13.7% in the range 5.5…6.5 ppm. Although this difference is still within the combined uncertainty of the instruments, it hampers the detailed investigation of trends in the lower stratospheric water vapor budget, which are of the same order of magnitude and highly uncertain, even in their sign (e.g. Hegglin et al., 2014; Lossow et al., 2018).

We investigate RHi measurements in cirrus clouds from AIMS and SHARC as an independent metric of the absolute accuracy of the $H_2O$ measurements. This is not straightforward, as RHi in cirrus clouds is known to differ significantly from saturation depending on the dynamics of the cloud. Still, considering a sufficiently large data base, the data can be used as an independent indicator of the absolute accuracy of the measurements under UTLS conditions. Data from both instruments have a mode value close to ice saturation (less than 10% difference of mean value for all flights). An overall instrumental or

sampling bias seems unlikely since flights with mean super- and subsaturation in clouds are almost evenly distributed. The same holds for a possible bias in the aircraft temperature measurement which would similarly propagate into the RHi distribution. However, we do observe a drift between the in-cloud measurements of the two instruments over the course of the measurement campaign. While AIMS measures higher RHi values than SHARC in the beginning of the campaign, mean RHi values agree much better during the second half of the campaign. When considering the entire data set (including clear

sky data), this drift is not apparent which makes a change in the performance of one instrument unlikely.

A comparison of the measured $H_2O$ mixing ratios with ECMWF IFS data is accomplished using the same methodology as for the instrument intercomparison. The gridded ECMWF data are interpolated in space and time along the flightpath of HALO with a resolution of 0.1 Hz. Measurement and model show generally good agreement throughout the upper troposphere with bin-wise mean values of the difference typically within ±10% (consistent with, e.g., Flentje et al. (2007))

with a slight tendency towards a model dry bias which, however, is not statistically significant. Below mixing ratios of 30 ppm, we observe a significant wet bias in the ECMWF model with highest mean deviation from the measurements around 6 ppm or at a potential temperature of 355 K, respectively. In that regime, mean deviations are in the order of 100% with an IQR of 70 to 140%. The large wet bias of the model in the tropopause region is consistent with findings in previous studies, e.g., by Kunz et al. (2014) or Dyroff et al. (2015). The model wet bias decreases substantially at higher potential

temperatures leading to a mean difference of only 17% at potential temperatures above 370 K. The fact that the model bias shows a clear maximum at the tropopause indicates that this issue is likely caused by a too weak humidity gradient at the tropopause rather than an overall bias of stratospheric mixing ratios. Kunz et al. (2014) found a similar feature with good agreement between FISH measurements and EMCWF reanalysis data at altitudes higher than 6 km above the tropopause. The issue of too weak gradients at the tropopause is discussed extensively, e.g., by Birner et al. (2002), Gray et al. (2014)



and subsequently by Saffin et al. (2017). In particular, the lower stratospheric wet bias is very sensitive to the horizontal interpolation of the specific humidity field in the semi-Langrangian IFS model (Diamantakis, 2014) leading to a too high diffusivity which in turn causes a cold bias at the extratropical tropopause (Stenke et al., 2008). However, it is difficult and cost intensive to address the issue in the model since it would need to adjust core dynamical model processes or increase the

model resolution, respectively (Saffin et al., 2017; Pope et al., 2001). Additionally, the model suffers from a lack of assimilated information on lower stratospheric water vapor since specific humidity data from radiosondes is only assimilated below a certain threshold pressure level (depending on the type of sonde, see Andersson et al. (2007)). Given the large model uncertainty in $H_2O$ concentrations close to the tropopause, it renders difficult to, e.g., correctly evaluate the radiative effects of water vapor in that region where the atmosphere is very sensitive to even small changes in $H_2O$ (Solomon et al., 2010;

Riese et al., 2012).

Despite the limitation to one-dimensional data for the in situ measurements, high spatial resolution data as obtained from aircraft can help to point out important small scale differences which are difficult to access when comparing model to satellite data due to their limited (especially vertical) resolution (e.g. Lamquin et al., 2009). The intercomparison shows, that our approach to compare in situ data with model data can be particularly useful to investigate model performance around the

tropopause. Hence, it could be worthwhile to extend this type of intercomparison to reanalysis data like the new climate reanalysis data set (ERA-5) of the ECMWF or include further NWP models like the Icosahedral Nonhydrostatic (ICON) model from the German Weather Service.

Data availability:

Data are accessible via the HALO data base (https://halo-db.pa.op.dlr.de/mission/2).

Competing interests:

Christiane Voigt and Ulrich Schumann are members of the editorial board of the joint ACP/AMT Special Issue "ML-CIRRUS – the airborne experiment on natural cirrus and contrail cirrus in mid-latitudes with the high-altitude long-range

research aircraft HALO".

Acknowledgements:

We thank the DLR flight department and Andreas Minikin for great support during the campaign and Klaus Gierens for helpful comments on the manuscript. Support by the Helmholtz Association under contract W2/W3-60 and by the German

science foundation within the DFG-SPP HALO 1294 via grant VO1504/4-1 (CV), JU3059/1-1 (TJ), KR 2957/1-1 (MK) and SCHI-872/2-2 (CR) is greatly acknowledged.



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





**Table 1 Measurement technique, range and uncertainty of the different instruments. Resolution values in brackets are time resolutions used for this intercomparison.**

| Instrument | Technique | Measured quantity | Range [ppm] | Resolution [s] | Uncertainty |
|---|---|---|---|---|---|
| **AIMS** | Mass spectrometry | gas phase $H_2O$ mixing ratio | 1 – 500 | 0.3 (1) | 7-15% |
| **FISH** | Lyman-$\alpha$ fluorescence | total $H_2O$ | 1 - 1000 | 1 | 6% $\pm$ 0.4 ppm |
| **SHARC** | TDL | gas phase $H_2O$ | 10 - 50000 | 1 | 5% $\pm$ 1 ppm |
| **HAI** (1.4 µm closed path channel) | TDL | total $H_2O$ | 20 - 40000 | 0.7 (1) | 4.3% $\pm$ 3 ppm |
| **WARAN** | TDL | total $H_2O$ | 100 - 40000 | 2.3 | 50 ppm or 5% |

5   **Table 2 Statistic summary of the five instruments including number of points entering the comparison, mean deviation and spread of the data.**

| Instrument | Number of data points | Mean deviation from reference [%] | Spread: Quartiles (10/90 percentiles) [%] |
|---|---|---|---|
| **AIMS** | 151947 | +1.4 | -2.2 / +5.3 (-5.8 / +9.5) |
| **FISH** | 94392 | -2.2 | -4.6 / +0.6 (-9.0 / +3.6) |
| **SHARC** | 149741 | -1.4 | -3.6 / +0.6 (-6.4 / +3.1) |
| **HAI** | 92277 | +2.3 | -0.4 / +3.1 (-2.1 / +6.4) |
| **WARAN** | 19550 | -7.5 | -11.3 / -1.7 (-20.3 / +4.1) |

**Table 3 Relative difference between ECMWF IFS data and measurements for different potential temperatures.**

| Potential Temperature Range [K] | # of points | Mean relative difference [%] | Standard deviation of relative difference [%] |
|---|---|---|---|
| > 370 | 761 | 16.9 | 8.9 |
| 360 - 370 | 1087 | 36.7 | 20.1 |
| 350 - 360 | 1759 | 87.5 | 49.1 |
| 340 - 350 | 1210 | 30.0 | 29.8 |
| 330 - 340 | 2213 | 11.3 | 25.1 |





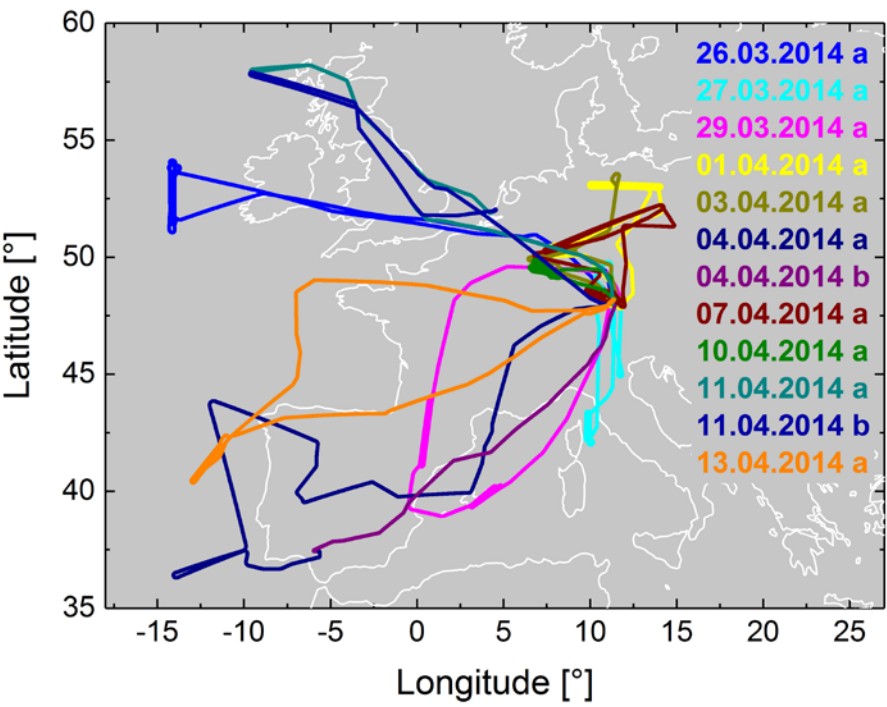

**Figure 1 Flight tracks of 12 research flights during the ML-CIRRUS campaign in March / April 2014 used for this study. Latitudes between 36°N and 57°N were covered mainly over Central and Western Europe.**





**Figure 2 Examplary water vapor molar mixing ratio measurements for the research flight on 3 April 2014. AIMS (black) and SHARC (green) measured in situ gas phase H₂O while FISH (blue), HAI (orange) and WARAN (red) measured total water. Panels (a) and (b) are profiles of H₂O in situ measurements plotted against potential temperature. Sequence (a) is the descent between 54904 s and 58522 s, Sequence (b) corresponds to the descent between 59139 s and 61398 s. Panel (c) is the time series of the complete flight including the HALO flight altitude in grey.**



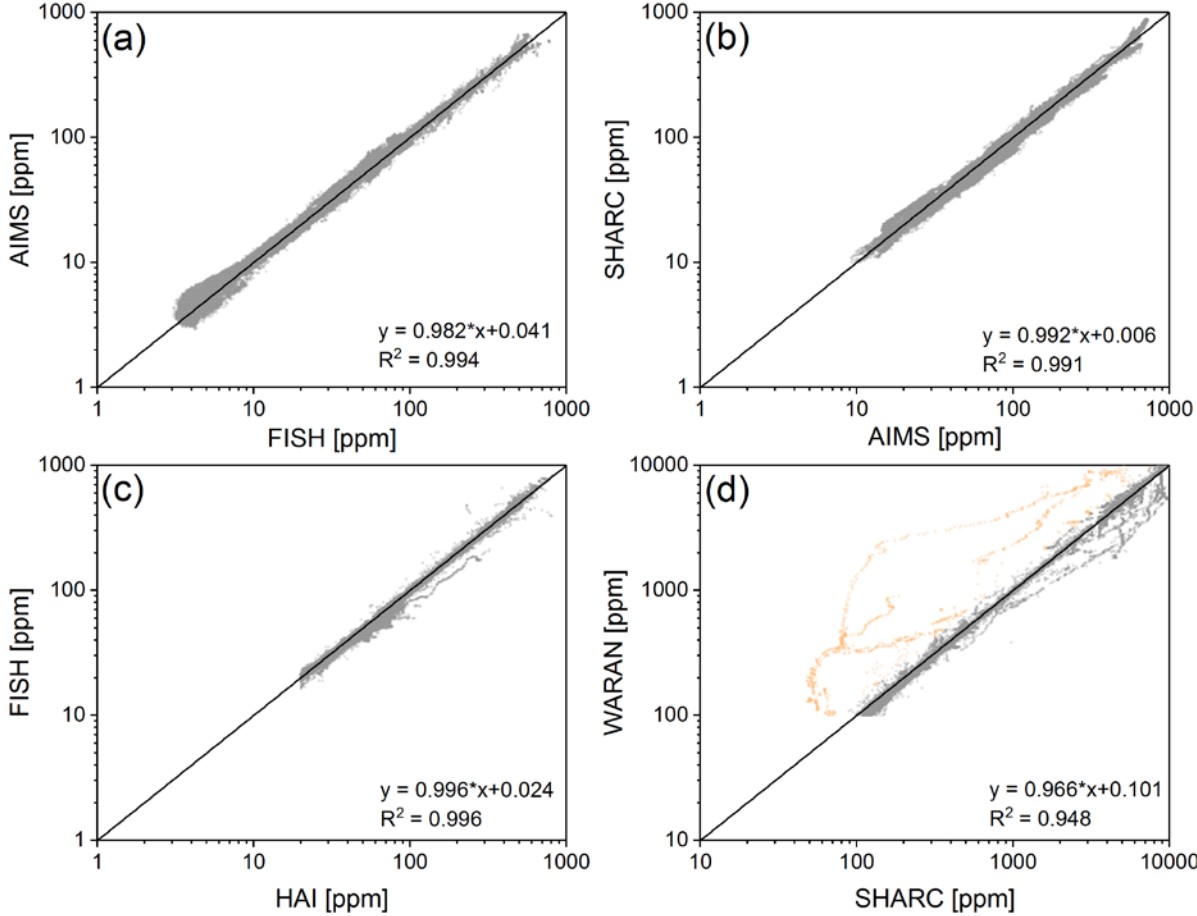

**Figure 3 Scatterplots of data from the five in situ water vapor instruments on HALO during ML-CIRRUS. (a) clear sky measurements of AIMS and FISH covering stratosphere and upper troposphere, (b) AIMS and SHARC measuring gas phase H$_2$O. This plot thus includes in-cloud gas phase H$_2$O data. (c) HAI vs FISH for clear-sky upper tropospheric mixing ratios. (d) WARAN vs SHARC data extending up to 10000 ppm with a lower cutoff of the WARAN at 100 ppm. The strong wet bias of the WARAN which occasionally occurs during the first ascend of the plane are marked orange. These data points are left out for the further intercomparison.**





**Figure**
**4 Relative difference of the measurements of AIMS, FISH, SHARC, HAI and WARAN from the mean H₂O molar mixing ratio**
**value which is used as reference (details see text). The small dots are the single measurement points (1 Hz values). The big squares,**
**triangle and circle are mean values of the relative difference for specific bins of H₂O mixing ratio. The broad bars represent the 25**
5 **and 75 percentile while the narrow bars stand for the 10 and 90 percentile within the bins. All points with a deviation between -1%**
**and +1% fall on the ±1 line. Values in the grey box on the left hand side represent the overall mean values for the different**
**instruments.**





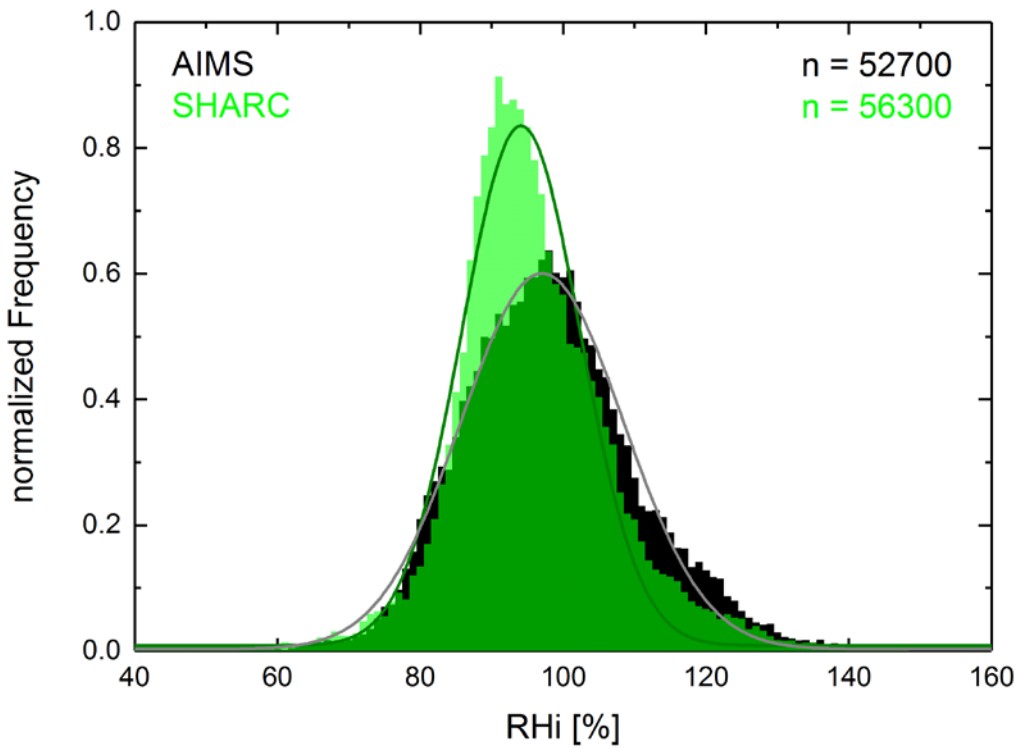

**Figure 5 PDFs of relative humidity with respect to ice calculated from AIMS (black) and SHARC (green) data and the static air temperature measurement on HALO inside cirrus clouds. Dark green indicates overlap regions. The cloud flag is the same used for filtering the total water measurements. The centre of the respective distribution is 94% for SHARC and 97% for AIMS.**




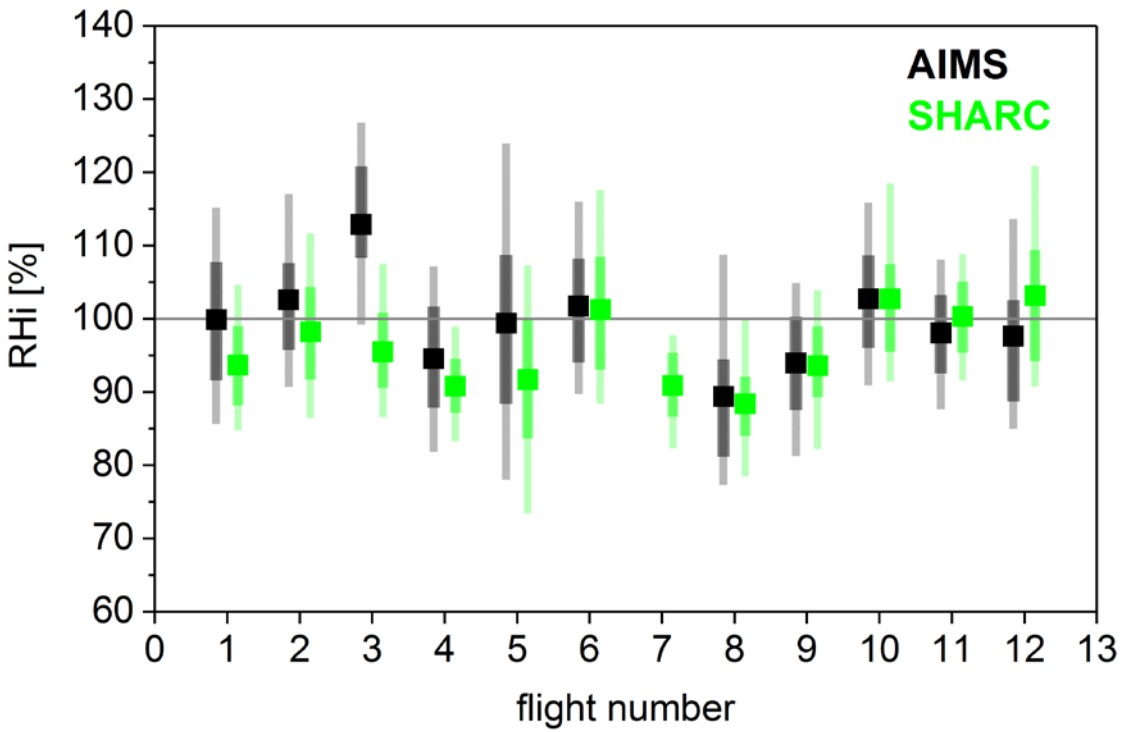

**Figure 6 Mean values for RHi inside cirrus measured by AIMS (black) and SHARC (green) for each ML-CIRRUS flight. Broad bars denote the interquartile range, narrow bars the 10/90 percentile range.**





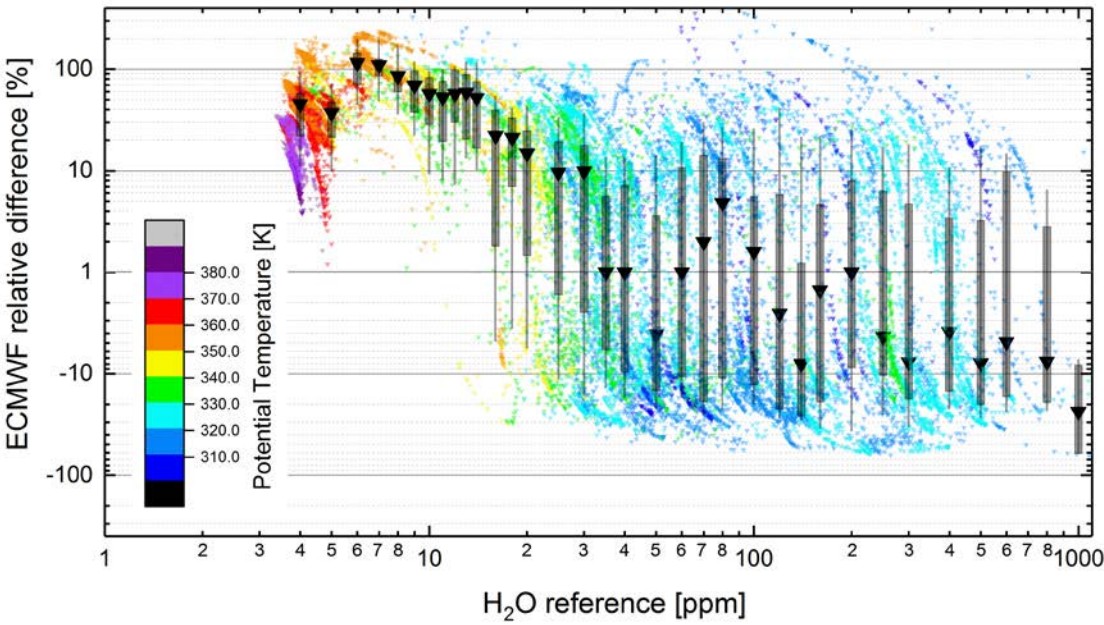

**Figure 7 Relative difference of the ECMWF analysis data interpolated for all ML-CIRRUS flights. Model data are interpolated in space and time on each flight track. The reference value on the x-axis is the same as in Figure 4. As in Figure 4, the triangles and bars represent the mean values, 25/75 and 10/90 percentiles, respectively. Single data points are colour coded with potential temperature.**