# Peer review of "Intercomparison of mid-latitude tropospheric and lower stratospheric water vapor measurements and comparison to ECMWF humidity data"

_Atmospheric Chemistry and Physics, 2018_

## Referee Comment (RC1) · Anonymous Referee #1 · 14 Aug 2018

Overview: This paper describes comparison results from a recent HALO aircraft campaign with a payload including multiple water vapour/total water instruments. The measurements are found to agree reasonably well both internally, and with output from an ECMWF forecast model. The publication provides a service to the community who would be interested in analyzing this data, but there is little in the way of scientific analysis in the paper as it stands. It may be better suited to AMT rather than ACP. It should be ultimately publishable, after addressing comments from all reviewers. Comments below with a * are the most substantial. The others are largely editorial.

Detailed comments:

[Figure]

Overall: The model comparison seems to be a small part of the paper, and perhaps doesn't need to be included in the title.

Page 1: line 17, change "turned out to be" to "is"

Page 2: line 11/12, rewrite sentence...does this not happen in a "non-changing climate"? also, eliminate the "like; e.g."

Page 2: line 12/14, rewrite this sentence " The radiative effect of clouds is much more complex than the effect of greenhouse gases due to very inhomogeneous cloud cover as well as microphysical and radiative properties of clouds at different altitudes." what does "more complex" really mean?

Page 2: line 14, change "countervailing" to "opposing"

*Page 2: line 17-26, What are you really trying to say here? I think it is: 1) It's hard to measure water vapor 2) the accuracy required depends on the research question 3) For radiative questions in the stratosphere, you want an accuracy of 1ppm or less 4) For cloud questions, you want an accuracy of 10% or less RHi

Rewrite to make those points clear, or possibly even delete the entire paragraph, as it is not critical for the main points in the paper.

*Page 2: line 28, is this 10% absolute water vapour measurement in the stratosphere? Or is it in the troposphere, or is it 10% RHI? I believe the campaigns mentioned imply 10% absolute and in the stratosphere, but if that is so, it should be made clearer in the text.

Page 2: line 33, instead of "was improved compared" perhaps instead "was better relative"

Page 3: line 11-13, rewrite sentence

Page 3: line 15, make measurement plural.

Page 4: line 4/5, sentence needs a verb

Page 4: line 14, change "utilized" to "use"

*Page 6: line 9, (HAI) Why do you only use data from one of the four channels?

Page 6: line 26, change " we use a couple of other parameters" to " we use additional parameters"

Page 6, line 30 change " of the Cloud" to "from the Cloud"

Page 7, line 5-9, this paragraph just needs to be rewritten, or even mostly deleted. Really, only the last sentence is needed.

Page 7: line 12, change "on the investigation" to " rather the investigation"

Page 7: line 12, change "consists" to "consist"

Page 7: line 24, change " vertical distance to the cirrus upper edge" to " vertical distance from the cirrus upper edge"

*Page 8: lines 13-20, You note that supersaturated cloud free cases are not considered. Do you have an idea of the fraction of the filtered data falls under this criterium? Does this bias your results in any way?

*Page 9: first paragraph, I'm not quite clear what your reference is. Is the lower stratosphere only using FISH and AIMS; and the upper troposphere clear sky, there are 4 instruments? Or, is the reference always the 4 instruments mentioned (without WARAN)?

Figure 2: caption, delete Examplary and just start with "Water Vapor"

Page 9: line 15, add a comma after "Except for the WARAN"

*Page 9: line 23, states " Sequences with such contamination are identified for the entire data set and filtered out for the intercomparison." Can you note how much data you had to throw out because of this contamination?

Figure 3: caption, change " WARAN which occasionally occurs during the first ascend" to " WARAN that occasionally occurs during the first ascent"

\*Page 10: line 19/21 states " As shown in Table 2, the mean deviations of AIMS, FISH, SHARC and HAI are below 2.5%, indicating that there is no consistent systematic bias in any single instrument." Although true when averaged over the entire range measured, that is not true for specific water vapor ranges. It may be instructive to make plot those via decade of water vapor. And, in fact you effectively contradict that statement with the one on line 26.

\*Page 10: line 26, I'm a little confused what you're trying to say with " In fact, there is a systematic difference between both instruments between 4 and 10 ppm." It sounds like the difference between the instruments is 4-10 ppm; but I think you really mean when the reference value is between 4 and 10 ppm. I suggest a rewrite.

\*Page 10: line 27/28, states "Interestingly, the difference between the instruments for the driest conditions (3.5 to 4.5 ppm) is smaller than for the next several bins (2.4% versus 6.5%)" Is this really a robust conclusion? There seems to be a large amount of spread there.

Page 11: line 16, missing space between "of" and "meteorological"

\*Page 11: discussion of RHi, have you considered the effect of uncertainty of the T & P measurements to the RHi? (looking further in the paper...if there is a temperature bias, what is the potential impact on the RHi distributions?)

\*Figure 5, since there are different numbers of points included, is there any bias for conditions when SHARC sampled and AIMS did not? Do you get the same picture if you only select the subset of points where both instruments measured at the same time?

\*Page 11: line 29-31, I assume the shift you're talking about here is the difference from a peak at 100%? (the 97% and 94% vs an expected value of 100%). If so, please

make that clear. Or are you talking about the shift in distribution between instruments.

*Page 12: first paragraph, is there a difference between flights in the average ambient temperature or pressure for each flight? Could that contribute to flight by flight differences (if the water instruments have some sensitivity to ambient conditions)?

Page 12: line 20; the paper here acknowledges temperature could be an issue, but what does "significantly off" mean. What temperature or pressure bias could cause a 6% change in RH at the conditions sampled?

*Page 13/14: comparison with model, I'm not sure that an interpolation the model to the 1 Hz aircraft locations is really the best way to do the comparison. It seems that that is attempting to impose structure on the model that is non-existent. Another tactic would be to average the aircraft points within the grid box of the model, and then do the comparison. It may produce essentially the same result, but reduce the scatter on the plot. This should probably also be considered with regards to vertical resolution as well. And, as noted elsewhere in the paper, whether the model and measurements are looking at the same height relative to the tropopause is also probably a factor. An attempt to quantify whether there is such a difference using the aircraft vertical profiles to identify a tropopause height, and then compare to the model representation would be useful in explaining discrepancies between model and measurements.

---

## Referee Comment (RC2) · Anonymous Referee #2 · 23 Aug 2018

Review of acp-2018-744:

Short Summary: The authors present an intercomparison of gas-phase (i.e. clear sky) airborne in situ water vapor measurements onboard the DLR research aircraft HALO during the mid-latitude ML-CIRRUS mission. This publication is important as the first comprehensive intercomparison of all the major research hygrometers of the German research community: HAI, SHARC, WARAN, AIMS, and FISH. Although the agreement of the hygrometers has improved significantly compared to studies from recent decades, systematic differences remain under specific meteorological conditions (differences on the order of 10% for mixing ratios below 10 ppm). The authors compare

the measurements to model data where we observe a model wet bias in the lower stratosphere close to the tropopause, likely caused by a blurred humidity gradient in the model tropopause.

Review: General Comments

This is an excellent manuscript and of significant interest to the water measurement community. The authors justify the importance of accurate water vapor measurements, and then carefully quantify the differences between state-of-the-art instruments. Since the focus of this paper is on intercomparison, and very brief on scientific analysis, this manuscript is more appropriate for AMT. Other key water intercomparison papers appear in AMT (e.g., Fahey et al., 2014). Subject to the other reviewers and the editor, I recommend that the authors submit this manuscript to AMT instead of ACP.

Specific Comments

1. Page 7, line 20: For measurements within clouds, how do you know that the relative humidity should be 100% with respect to ice? Anvil ice are likely close to ice saturation, but there is much literature that other ice clouds are expected to be supersaturated (for instance, upper parts of ice clouds and ice clouds forming in-situ). The asymmetric tails of relative humidity toward higher supersaturation (e.g. Figure 5, Figure 6 and page 11) are evidence of supersaturated environments.

2. In the data analysis, is it possible for you take into consideration where (vertically) in the cloud the measurements were made?

3. 3a. Which flight number is used for Figure 5? The discussion on page 12 identifies specific flights where supersaturation is expected. 3b. Were these flight segments (e.g. high updraft velocity) excluded from Figure 5 and the relative humidity analysis?

4. Page 9, line 13: Sahara dust is mentioned twice in the manuscript (e.g. pages 9 and 14). What is the significance of Saharan dust to humidity measurements?

5. Page 15, lines 31-32: this manuscript concludes that ECMWF model bias is due

to a too-small humidity gradient across the tropopause. I recommend that the authors add a figure that shows example vertical profiles of $H_2O$ from model and aircraft, with the tropopause height labelled, to demonstrate this gradient.

Technical Corrections Below are minor editing comments,

1. Page 1, line 18: change "number relevant scientific questions" to "number of relevant scientific questions"

2. When pointing to a section, capitalize "Section", examples on page 3, line 20, and page 4, line 1.

3. Page 8, line 2: remove double period.

4. Page 14, line 27: change "cloud" to "clouds".

---

## Referee Comment (RC3) · Anonymous Referee #3 · 28 Aug 2018

The manuscript describes an intercomparison of water vapor instruments aboard the DLR HALO aircraft during the 2014 ML-CIRRUS airborne campaign. The manuscript is generally very well written. I recommend that some relatively minor changes be made prior to publication. Recommended changes follow, in order of more important to less important.

Section 4.2: This section discusses the filtering of campaign data from the five water vapor sensors for the purpose of enabling the intercomparison. This is an important activity, and the methods by which it is done can have measurable effects on the outcome. The primary utilities of an intercomparison are (1) to create a unified, selfconsistent dataset which enables greater scientific meaning than would be obtained using only one measurement; and (2), to create a means to understand data obtained when the intercompared sensors are operating without the other(s). This intercomparison generally falls into the second category. For that purpose, most of the data filtering described seems appropriate, but in the final paragraph, a process is described which throws away data for which there are explainable or unexplainable problems. These data, if they appear in the official project data archive, should not be removed because they disagree with the other measurements. It is in these disagreements that one can learn about the ultimate reliability of a measurement, and removing these data hides that information. If these data are actually not in the data archive because the suppliers of the data had already marked it as unreliable, then this fact should be stated.

Section 5: The final two sentences of this section (page 9, lines 20-24) describe additional removal of data due to the disagreement shown. This is similar to the issue described above. Again, if the data are in the archive, the comparison should include those data. If they are not, the authors should note that.

Section 3.1: The AIMS instrument is described. On page 5, line 2, the text indicates that the instrument "was calibrated once or twice during each research flight." How consistent were the in-flight calibrations, both in a single flight, and among all the flights? At what conditions were they performed? Is there any trend to the differences in calibrations?

Section 3.4: The HAI instrument is described, including the fact that it uses two different wavelengths. But on page 6 line 9, the statement is made that only the 1.37 um data are used in this intercomparison. Why is that?

Section 3.x: These sections describe each of the instruments, and provide some information on accuracy and calibration. Unfortunately, the same information isn't provided for all of the instruments. The authors should amend each of the sections to include all of the same important information, including accuracy, precision, time response, and

method/timing of calibrations. Some, but not all, of this information is in Table 1.

Section 1: This section provides background on airborne water vapor measurements and intercomparisons done with those measurements, including ground-based inter-comparisons. The authors might also include intercomparisons reported by Jensen of measurements made during the NASA ATTREX campaigns on the Global Hawk air-craft in the UT/LS/TTL. Comparisons during ATTREX were generally better than those from AquaVIT-1 and MACPEX.

Section 4.3: This section describes the selection of the reference value, and mentions the fact that no single instrument covers the entire range of values observed. This seems to imply that it would be common for some combination or combinations of instruments to be used on this and other German aircraft during other campaigns. Is that the case? If so, which instruments typically fly together? And how to they generally compare in the ranges where they have overlapping measurements?

Section 6: On page 15, line 20, drift is discussed, but the statement is made that observed relative changes between measurements made by the AIMS and SHARC instruments are not due to drifts in either instrument. As this seems to be difficult to reconcile with the observations, what do the authors suggest is the cause or explanation?

Minor word changes, etc.: Page 1 Line 16 – suggest replacing "turned out to be" with "is" Line 24 – suggest replacing "total mean values even agree" with "and total mean values agree" Line 31 – suggest replacing "deficit" with "error"

Page 2 Line 9 – suggest removing "their" Line 32 – suggest adding "but as-yet undocu-mented" before "campaigns" Line 33 – suggest adding "during AquaVIT-1" before "was improved" and replacing "compared" with "relative"

Page 3 Line 15 – suggest replacing "major" with "primary" here and elsewhere.

Page 7 Line 25 – "less" should be replaced by "fewer"

Page 10 Line 5 – suggest replacing "way" with "well"

Page 11 Line 16 – typographical error: "ofmeteorological" should be "of meteorological"

Page 13 Line 14 – should "interpolated" be "averaged" ?

Page 14 Line 23 – add comma after "hygrometer"

Page 15 Line 28 – replace "IQR" with "interquartile range"

Page 16 Line 12 – replace "access" with "assess"
* * *

---

## Author Comment (AC1) · 11 Oct 2018

**Overview:**

This paper describes comparison results from a recent HALO aircraft campaign with a payload including multiple water vapour/total water instruments. The measurements are found to agree reasonably well both internally, and with output from an ECMWF forecast model. The publication provides a service to the community who would be interested in analyzing this data, but there is little in the way of scientific analysis in the paper as it stands. It may be better suited to AMT rather than ACP. It should be ultimately publishable, after addressing comments from all reviewers. Comments below with a * are the most substantial. The others are largely editorial.

*We thank the reviewer for the constructive comments which are addressed below. Since the paper focuses on the atmospheric water vapor distribution and the instrument performance under real atmospheric conditions rather than discussing measurement techniques, we would highly appreciate if it is considered for ACP. In that sense, we see it as a continuation of studies like Rollins et al. (JGR, 2014), Kiemle et al. (ACP, 2008), Dyroff et al. (Q. J. R. Meteorol. Soc., 2015) and Kunz et al. (ACP, 2014) where different sets of airborne $H_2O$ measurements are intercompared to each other and to NWP model data, respectively. Similar to those studies, we go beyond a simple discussion of the different measurement techniques. Motivated by the comment of reviewer #2, we further extended the discussion of the ECMWF intercomparison with an additional figure supporting our interpretation of too weak humidity gradient in ECMWF above the tropopause.*

**Specific comments:**

Overall: The model comparison seems to be a small part of the paper, and perhaps doesn't need to be included in the title.

*Despite the model intercomparison being a small part, it is an essential part of the work. Thus we would appreciate to have that part included in the title to also address the NWP model community.*

*Page 2: line 17-26, What are you really trying to say here? I think it is: 1) It's hard to measure water vapor 2) the accuracy required depends on the research question 3) For radiative questions in the stratosphere, you want an accuracy of 1ppm or less 4) For cloud questions, you want an accuracy of 10% or less RHi. Rewrite to make those points clear, or possibly even delete the entire paragraph, as it is not critical for the main points in the paper.

*We fully agree with the summary and excuse the lack of clarity of the paragraph. Therefore we rewrote the paragraph according to the reviewer's suggestions and shortened it significantly to make it more concise.*

*Page 2: line 28, is this 10% absolute water vapour measurement in the stratosphere? Or is it in the troposphere, or is it 10% RHI? I believe the campaigns mentioned imply 10% absolute and in the stratosphere, but if that is so, it should be made clearer in the text.

*Meant is here the relative uncertainty of the water vapor mixing ratio in both upper troposphere and lower stratosphere. Of course, this uncertainty also translates to an uncertainty of RHi in the same order of magnitude in the upper troposphere. We changed the text to make that clearer:*

*"…that the relative measurement uncertainty in water vapor mixing ratio was significantly higher than 10%, even occasionally exceeding 100% in at the lowest mixing ratios in the lower stratosphere…"*

*Page 6: line 9, (HAI) Why do you only use data from one of the four channels?

The 1.37 µm closed path channel is the most robust one and, unfortunately, the instrument experienced technical issues with the other three channels. In consequence, only that channel could provide data within the required uncertainty margin. A brief statement is added to the manuscript:

*"For this work, we use data from the 1.37 µm closed cell channel of HAI in the range of 20 to 40000 ppm since only that channel provided data within the required uncertainty margin during ML-CIRRUS."*

*Page 8: lines 13-20, You note that supersaturated cloud free cases are not considered. Do you have an idea of the fraction of the filtered data falls under this criterium? Does this bias your results in any way?

*In terms of upper tropospheric gas phase $H_2O$ measurements, SHARC provides the most comprehensive data set since AIMS data exhibit calibration and instrument zeroing gaps. From this data set, we have 6647 points with supersaturation with respect to ice in conditions labelled cloud free. That is around 6% of all cloud free data points measured by SHARC or around 4% of the entire SHARC data set (please note that SHARC only measures in the troposphere).*

*Since none of the instruments directly measures relative humidity, we do not expect that supersaturated data are biased towards the rest of the data set. All data shown here are measured in heated (and temperature controlled) conditions, hence the relative humidity in the measurement cells is always way below saturation which allows to exclude condensation effects.*

*Page 9: first paragraph, I'm not quite clear what your reference is. Is the lower stratosphere only using FISH and AIMS; and the upper troposphere clear sky, there are 4 instruments? Or, is the reference always the 4 instruments mentioned (without WARAN)?

*When there are valid data points from at least two instruments for a single time step, we use all available data points (between 2 and 4) to calculate the mean value which is used as reference.*

*In the lower stratosphere, only data from FISH and AIMS are available. Hence, the reference value there is simply the mean value of these two instruments. In the upper troposphere, generally all four instruments are used for the calculation of the reference except for time steps where one or two instruments have experienced measurement gaps.*

*We added a sentence to make that clearer:*

*"For the lower stratosphere, the reference is the mean value of AIMS and FISH measurements. For the troposphere, generally all four instruments are used for the calculation of the reference except for cloud sequences and depending on data availability."*

*Page 9: line 23, states " Sequences with such contamination are identified for the entire data set and filtered out for the intercomparison." Can you note how much data you had to throw out because of this contamination?

*The rejected amount of data differs for both instruments since the WARAN seemed to be more susceptible to such contamination compared to HAI (see last part of Figure 2). In total, sequences with liquid cloud encounters followed by dry condition like in Figure 2 occurred only very rarely, the filtering due to that kind of contamination affected less than 1% of the data which is now also stated in the manuscript.*

*Page 10: line 19/21 states " As shown in Table 2, the mean deviations of AIMS, FISH, SHARC and HAI are below 2.5%, indicating that there is no consistent systematic bias in any single instrument." Although true when averaged over the entire range measured, that is not true for specific water vapor ranges. It may be instructive to make plot those via decade of water vapor. And, in fact you effectively contradict that statement with the one on line 26.*

*That is exactly what we are trying to say: When looking at the entire data set, there seems to be no significant deviation between the measurements. However, when looking more closely, systematic deviations occur in certain humidity ranges and those are analyzed in detail in the following paragraph. We reformulated the sentence to make the statement more clear:*

*"…indicating that there is no consistent systematic bias when averaging over the entire data set."*

*Page 10: line 26, I'm a little confused what you're trying to say with " In fact, there is a systematic difference between both instruments between 4 and 10 ppm." It sounds like the difference between the instruments is 4-10 ppm; but I think you really mean when the reference value is between 4 and 10 ppm. I suggest a rewrite.*

*Sorry, that sentence is confusing. We rewrote it to:*

*"In fact, there is a systematic difference between both instruments for humidity conditions between 4 and 10 ppm."*

*Page 10: line 27/28, states "Interestingly, the difference between the instruments for the driest conditions (3.5 to 4.5 ppm) is smaller than for the next several bins (2.4% versus 6.5%)" Is this really a robust conclusion? There seems to be a large amount of spread there.*

*There is indeed a large amount of spread in the data. We do see a difference but it is questionable if this difference is statistically significant; probably not. However, we do observe that the deviations between the instruments are both positive and negative for the driest bin while they show a more consist direction for the bins above.*

*We added a sentence to clarify that we do not judge the statistical significance:*

*"However, the spread in data points is too large to judge if this difference is significant."*

*Page 11: discussion of RHi, have you considered the effect of uncertainty of the T & P measurements to the RHi? (looking further in the paper...if there is a temperature bias, what is the potential impact on the RHi distributions?)*

*The uncertainty of the temperature and pressure measurement onboard HALO is 0.5 K and 0.3 hPa, respectively. Assuming a bias in temperature in that order of magnitude, RHi values in the upper troposphere would be shifted by 7 to 8%, depending on the absolute temperature. The uncertainty in*

*pressure hardly affects RHi. A bias in the pressure measurement of 0.3 hPa would shift RHi values by around 0.2%.*

*Figure 5, since there are different numbers of points included, is there any bias for conditions when SHARC sampled and AIMS did not? Do you get the same picture if you only select the subset of points where both instruments measured at the same time?

*The number of points for AIMS is lower due to measurement gaps during routinely performed calibration and zeroing procedures. These procedures do not correlate with certain atmospheric conditions or events, hence we do not expect a bias there. Plotting only data from sequences where both instruments measured, the picture looks basically identical.*

*Page 11: line 29-31, I assume the shift you're talking about here is the difference from a peak at 100%? (the 97% and 94% vs an expected value of 100%). If so, please make that clear. Or are you talking about the shift in distribution between instruments.

*Yes, the shift relative to the expected saturation is meant here. We rewrote the sentence to make that clear:*

*"However, the question remains whether the slight shift of the centre of the RHi distribution relative to 100% is caused by systematic instrument biases…"*

*Page 12: first paragraph, is there a difference between flights in the average ambient temperature or pressure for each flight? Could that contribute to flight by flight differences (if the water instruments have some sensitivity to ambient conditions)?

*This is an interesting question, however not straight forward to answer since we always flew multiple sampling legs on different altitude (and thus pressures) in order to sample in and above the clouds. The pressure levels where we found clouds were very similar over the campaign and are similarly probed during the flights. Thus, a pressure bias should be leveled out in the flight by flight intercomparison.*

*Temperature conditions during the campaign were less uniform and, similar to pressure, there is a large spread in temperatures within each flight due to the changing flight altitudes. Looking at average cirrus temperatures, the warmest flight was flight number 10, the coldest one was flight number 12. Regarding RHi distribution and difference between AIMS and SHARC in Figure 6, they look very similar. From that, we see no indication that ambient temperature influences the RHi measurements. Looking e.g. at flight number 3 with the large spread between the instruments, ambient conditions were very average.*

*Page 13/14: comparison with model, I'm not sure that an interpolation the model to the 1 Hz aircraft locations is really the best way to do the comparison. It seems that that is attempting to impose structure on the model that is non-existent. Another tactic would be to average the aircraft points within the grid box of the model, and then do the comparison. It may produce essentially the same result, but reduce the scatter on the plot. This should probably also be considered with regards to vertical resolution as well. And, as noted elsewhere in the paper, whether the model and measurements are looking at the same height relative to the tropopause is also probably a factor. An attempt to quantify whether there is such a difference using the aircraft vertical profiles to identify a

tropopause height, and then compare to the model representation would be useful in explaining discrepancies between model and measurements.

*We agree that there are multiple ways to approach that intercomparison and it is clear that we obtain a rather scattered signal when comparing the variable 0.1 Hz measurements (spatial resolution of ~ 2.5 km) with the smoother model field. Here we tried to find a good compromise between eliminating the very small scale fluctuations and still keeping the characteristics of the measurement signal. Doing so, we can try to get an idea to what extend real variability can be represented in the model. We rewrote the description and interpretation of Figure 7 to make clear that the larger scatter of the single points is an expected feature due to resolution differences between measurement and model.*

*Concerning the conclusion that can be drawn from the intercomparison, the resolution issue obviously has to be kept in mind. However, the main conclusion of good agreement in the upper troposphere and a model wet bias in the lower stratosphere would also remain stable for longer averaging periods. Concerning vertical levels, the majority of measurement points are gathered on constant flight legs where an interpolation of the model data between adjacent levels seems the most appropriate approach.*

*Comparing the tropopause height between aircraft measurements and model would indeed be very interesting. Unfortunately, the profiles do not provide sufficient data to derive a measured tropopause height since we usually crossed the tropopause only a few times during a flight on very different horizontal positions. Furthermore, the meteorological conditions that were targeted during the campaign usually came along with rather complex tropopause structures. Hence, it turned out that potential temperature provides a more consistent picture in Figure 7 than using the calculated vertical distance to the ECMWF tropopause.*

**Technical comments:**

Page 1: line 17, change "turned out to be" to "is"

*changed*

Page 2: line 11/12, rewrite sentence...does this not happen in a "non-changing climate"? also, eliminate the "like; e.g."

*changed*

Page 2: line 12/14, rewrite this sentence " The radiative effect of clouds is much more complex than the effect of greenhouse gases due to very inhomogeneous cloud cover as well as microphysical and radiative properties of clouds at different altitudes." What does "more complex" really mean?

*rewritten*

Page 2: line 14, change "countervailing" to "opposing"

*changed*

Page 2: line 33, instead of "was improved compared" perhaps instead "was better relative"

*changed*

Page 3: line 11-13, rewrite sentence

*We tried to make the statement mor specific and changed it to:*

"Referring to the accuracy required to address the questions noted above, it seems that significant progress has been made in recent years. However, the current measurement accuracy still limits our ability to appropriately assess questions like e.g. stratospheric water vapor trends. "

Page 3: line 15, make measurement plural.

*changed*

Page 4: line 4/5, sentence needs a verb

*changed*

Page 4: line 14, change "utilized" to "use"

*changed*

Page 6: line 26, change " we use a couple of other parameters" to " we use additional parameters"

*changed*

Page 6, line 30 change " of the Cloud" to "from the Cloud"

*changed*

Page 7, line 5-9, this paragraph just needs to be rewritten, or even mostly deleted. Really, only the last sentence is needed.

*rewritten*

Page 7: line 12, change "on the investigation" to " rather the investigation"

*changed*

Page 7: line 12, change "consists" to "consist"

*changed*

Page 7: line 24, change " vertical distance to the cirrus upper edge" to " vertical distance from the cirrus upper edge"

*changed*

Figure 2: caption, delete Examplary and just start with "Water Vapor"

*changed*

Page 9: line 15, add a comma after "Except for the WARAN"

*changed*

Figure 3: caption, change "WARAN which occasionally occurs during the first ascend" to " WARAN that occasionally occurs during the first ascent"

*changed*

Page 11: line 16, missing space between "of" and "meteorological"

*changed*

**Anonymous Referee #2**

Short Summary: The authors present an intercomparison of gas-phase (i.e. clear sky) airborne in situ water vapor measurements onboard the DLR research aircraft HALO during the mid-latitude ML-CIRRUS mission. This publication is important as the first comprehensive intercomparison of all the major research hygrometers of the German research community: HAI, SHARC, WARAN, AIMS, and FISH. Although the agreement of the hygrometers has improved significantly compared to studies from recent decades, systematic differences remain under specific meteorological conditions (differences on the order of 10% for mixing ratios below 10 ppm). The authors compare the measurements to model data where we observe a model wet bias in the lower stratosphere close to the tropopause, likely caused by a blurred humidity gradient in the model tropopause.

**Review: General Comments**

This is an excellent manuscript and of significant interest to the water measurement community. The authors justify the importance of accurate water vapor measurements, and then carefully quantify the differences between state-of-the-art instruments. Since the focus of this paper is on intercomparison, and very brief on scientific analysis, this manuscript is more appropriate for AMT. Other key water intercomparison papers appear in AMT (e.g., Fahey et al., 2014). Subject to the other reviewers and the editor, I recommend that the authors submit this manuscript to AMT instead of ACP.

*We thank the reviewer for the constructive comments on the manuscript. We greatly acknowledge the important work done by the water vapor comunity during the AquaVIT laboratory experiments which are reported in Fahey (AMT,2014). However, as stated in the response to reviewer #1, we see this study in the line of airborne measurements and model intercomparisons like Rollins et al. (JGR, 2014), Kiemle et al. (ACP, 2008), Dyroff et al. (Q. J. R. Meteorol. Soc., 2015) and Kunz et al. (ACP, 2014) where different sets of airborne $H_2O$ measurements are intercompared to each other and to NWP model data, respectively. Regarding the section of RHi measurements in cirrus (compare e.g., Ovarlez (GRL, 2002)) and the model intercomparison, we think that our analysis goes beyond a discussion of different measurement techniques*

*Following the useful suggestion from the referee to include a figure showing humidity profiles of measurements and ECMWF, we further extended the discussion of the measurement-model intercomparison. The figure supports our interpretation of the general intercomparison that the systematic difference between ECMWF and the measurements in the lower stratosphere is mainly caused by a too weak humidity gradient of the model extending from the tropopause up to a few kilometers above.*

**Specific Comments:**

1. Page 7, line 20: For measurements within clouds, how do you know that the relative humidity should be 100% with respect to ice? Anvil ice are likely close to ice saturation, but there is much literature that other ice clouds are expected to be supersaturated (for instance, upper parts of ice clouds and ice clouds forming in-situ). The asymmetric tails of relative humidity toward higher supersaturation (e.g. Figure 5, Figure 6 and page 11) are evidence of supersaturated environments.

*We agree that we would not expect a symmetric distribution of RHi around saturation, which is discussed in more detail on page 11 and evident from Figure 5. Still, we expect the mode value of the distribution to be close to 100% assuming a sufficiently representative data set. We changed the sentence on page 7 accordingly to avoid confusion and added two exemplary references:*

*"Since we expect the mode value of the RHi distribution to be close to 100% inside cirrus (e.g., Ovarlez et al., 2002; Jensen et al., 2017b), this allows for an independent check of the absolute values of the gas phase water vapor measurements."*

2. In the data analysis, is it possible for you take into consideration where (vertically) in the cloud the measurements were made?

*This would indeed be a very interesting information, since one could expect a tendency to supersaturation at the upper edge of the clouds and vice versa for sublimation regions at the cloud base. However, the vertical position of the aircraft relative to the cloud is difficult to quantify from in-situ measurements since we have no information about the regions above and below the aircraft. From our point of view, the most promising approach to get information on cloud top an base heights is to use remote sensing methods like radar or LIDAR, the latter was also part of the ML-CIRRUS instrumentation (see Urbanek et al.,2017). However, a direct link between LIDAR and in-situ measurements is difficult due to the time lags between LIDAR and in-situ legs and the usually inhomogeneous structure of the cirrus clouds.*

3. 3a. Which flight number is used for Figure 5? The discussion on page 12 identifies specific flights where supersaturation is expected. 3b. Were these flight segments (e.g. high updraft velocity) excluded from Figure 5 and the relative humidity analysis?

*Figure 5 shows all available in-cloud data from the campaign. We expect the discussed sequences with high updraft velocity to contribute to the mentioned asymmetric tail towards supersaturation in Figure 5. We changed the figure caption to clarify the data basis.*

4. Page 9, line 13: Sahara dust is mentioned twice in the manuscript (e.g. pages 9 and 14). What is the significance of Saharan dust to humidity measurements?

*We do not expect any direct influence of Saharan dust on the in situ measurements but investigated a four day dust outbreak event along with the comparison to the ECMWF data. The question was whether the dust outbreak could have a negative impact on the model performance and could thus explain parts of the differences we observe between model and measurement. We only shortly discussed that on page 14 since we did not observe a substantial difference in the comparison between days with and without influence from Saharan dust.*

5. Page 15, lines 31-32: this manuscript concludes that ECMWF model bias is due to a too-small humidity gradient across the tropopause. I recommend that the authors add a figure that shows example vertical profiles of H2O from model and aircraft, with the tropopause height labelled, to demonstrate this gradient.

*We thank the referee for this useful suggestion. We added a Figure including two water vapor and temperature profiles from measurements and model to demonstrate the difference in humidity gradients. In both profiles, the thermal tropopause is well represented by the model. Both profiles exhibit the described features of (1) good agreement in the upper troposphere, (2) too weak humidity*

*gradient in the model directly above the tropopause, (3) Convergence of measurement and model above a certain distance to the tropopause.*

[Figure]

**Figure 8 Profiles of water vapor mixing ratio and temperature from in-situ measurements and ECMWF model. The blue line is the water vapor reference value from in-situ observations, the green line is the interpolated ECMWF model data. Data shown here originate from one ascent (a) and one descent (b) through the tropopause on April 11 2014 (flight #11). The water vapor profiles agree well in the upper troposphere, in the lower stratosphere we observe a stronger gradient in the measurements compared to the model. The vertical position of the thermal tropopause (black: measured by HALO, gray: ECMWF) is well represented in the model.**

*Accordingly, we added a paragraph to Section 5.4:*

*"The maximal differences close to tropopause mixing ratios indicate that the difference between measurement and model is caused by a too weak humidity gradient at the tropopause, partially explained by the model grid resolution of about 300 m vertically near the tropopause. Here, narrow inversions may form between subsiding dry stratospheric air and upward mixing of humid cold tropospheric air (Birner et al., 2002) which might not be covered by the coarse resolution of a global model. The difference in humidity gradients is directly evident in the humidity profiles. Figure 8 shows one ascent (a) and one descent (b) through the entire tropopause region on 11 April 2014. Consistent with Figure 7, we observe a good agreement between model and measurement in the troposphere. Directly above the tropopause, the humidity gradient in the model is weaker compared to the measurements for both profiles, resulting in overestimation of water vapor by the model in that region. This feature is independent from the absolute height of the tropopause (~11.8 km in (a), ~10.4 km in (b)), which is well represented in the model when comparing measured and modelled temperature profiles. With increasing vertical distance to the tropopause, measurement and model approach similar values, which is consistent with the overall intercomparison in Figure 7. The region above the tropopause where we observe a significant difference between measurements and model varies from around 1 km above the tropopause in Figure 8 (a) to around 3 km in (b). Thus, the weaker gradient is certainly no artefact of the vertical interpolation of the model."*

**Technical Corrections**

1. Page 1, line 18: change "number relevant scientific questions" to "number of relevant scientific questions"

*changed*

2. When pointing to a section, capitalize "Section", examples on page 3, line 20, and page 4, line 1.

*changed*

3. Page 8, line 2: remove double period.

*changed*

4. Page 14, line 27: change "cloud" to "clouds".

*changed*

**Anonymous Referee #3**

The manuscript describes an intercomparison of water vapor instruments aboard the DLR HALO aircraft during the 2014 ML-CIRRUS airborne campaign. The manuscript is generally very well written. I recommend that some relatively minor changes be made prior to publication. Recommended changes follow, in order of more important to less important.

*We thank the reviewer for the positive comments on the manuscript. Remarks and corrections are addressed individually in the following sections.*

Section 4.2: This section discusses the filtering of campaign data from the five water vapor sensors for the purpose of enabling the intercomparison. This is an important activity, and the methods by which it is done can have measurable effects on the outcome. The primary utilities of an intercomparison are (1) to create a unified, self-consistent dataset which enables greater scientific meaning than would be obtained using only one measurement; and (2), to create a means to understand data obtained when the intercompared sensors are operating without the other(s). This intercomparison generally falls into the second category. For that purpose, most of the data filtering described seems appropriate, but in the final paragraph, a process is described which throws away data for which there are explainable or unexplainable problems. These data, if they appear in the official project data archive, should not be removed because they disagree with the other measurements. It is in these disagreements that one can learn about the ultimate reliability of a measurement, and removing these data hides that information. If these data are actually not in the data archive because the suppliers of the data had already marked it as unreliable, then this fact should be stated.

Section 5: The final two sentences of this section (page 9, lines 20-24) describe additional removal of data due to the disagreement shown. This is similar to the issue described above. Again, if the data are in the archive, the comparison should include those data. If they are not, the authors should note that.

*We agree that a disagreement between data sets by itself is no reason to remove data from the intercomparison. What we are trying to do here, is to identify instrumental issues which would probably not be noticed when only flying on single instrument. The reasons to dismiss data caused by the three different effects (Section 4.2: AIMS pressure regulation, time shift; Section 5: liquid water contamination) need to be considered separately:*

- *The issue of the response time of the AIMS pressure regulation came up when comparing data from the different instruments. Data which are affected by that are also deleted from the archive which is now also stated in the manuscript. We are further working on a faster pressure regulation for the instrument.*
- *The occasional variable time shifts between different instruments can be treated in two ways for the intercomparison. One could either manually correct the time shift and include the data, or, as we decided to do, dismiss these data from the intercomparison. Simply including them without any correction would result in rather large artificial differences between the instruments which are not connected to the instrument performance itself. When flying only a single instrument, these time lags would probably barely be noticeable, except when*

*combining it with e.g. temperature time series to calculate relative humidity. From our experience, time lags between different sensors for temperature and humidity have to be considered individually for each aircraft setup.*

- *The issue of liquid water contamination in total water vapor instruments is more a sampling issue than an issue of instrument performance. From the instruments point of view, the measured value is not wrong, so we decided to exclude those data from the evaluation of the instrument performance. However, the lesson learned is, that one needs to be aware of contamination in total water instruments when measuring low $H_2O$ mixing ratios after flying through a liquid cloud. In cases where we are aware of such contamination, the general recommendation is to remove those data from the archive as we did for our own data.*

Section 3.1: The AIMS instrument is described. On page 5, line 2, the text indicates that the instrument "was calibrated once or twice during each research flight." How consistent were the in-flight calibrations, both in a single flight, and among all the flights? At what conditions were they performed? Is there any trend to the differences in calibrations?

*The scatter of calibration coefficients derived from the inflight calibration was very small (a few percent) and did not show a trend throughout the ML-CIRRUS campaign. Conditions and details of the inflight calibrations are discussed extensively in Kaufmann et al. (2016) where we show data from the exact same campaign to evaluate the performance of AIMS including a detailed description of the inflight calibration during ML-CIRRUS.*

Section 3.4: The HAI instrument is described, including the fact that it uses two different wavelengths. But on page 6 line 9, the statement is made that only the 1.37 um data are used in this intercomparison. Why is that?

*The 1.37 µm closed path channel is the most robust one and, unfortunately, the instrument experienced technical issues with the other three channels. In consequence, only that channel could provide data within the required uncertainty margin. A brief statement is added to the manuscript:*

*"For this work, we use data from the 1.37 µm closed cell channel of HAI in the range of 20 to 40000 ppm since only that channel provided data within the required uncertainty margin during ML-CIRRUS."*

Section 3.x: These sections describe each of the instruments, and provide some information on accuracy and calibration. Unfortunately, the same information isn't provided for all of the instruments. The authors should amend each of the sections to include all of the same important information, including accuracy, precision, time response, and method/timing of calibrations. Some, but not all, of this information is in Table 1.

*In this work, we tried to summarize the key parameters of each instrument which are important for the interpretation of the intercomparison. For FISH, HAI and AIMS there are dedicated instrument papers which describe calibration procedures (if applicable), sources of uncertainty and other instrument-specific issues. We added some information on the ground reference for the calibration of SHARC and WARAN.*

Section 1: This section provides background on airborne water vapor measurements and intercomparisons done with those measurements, including ground-based intercomparisons. The authors might also include intercomparisons reported by Jensen of measurements made during the

NASA ATTREX campaigns on the Global Hawk aircraft in the UT/LS/TTL. Comparisons during ATTREX were generally better than those from AquaVIT-1 and MACPEX.

*We are aware that there are a couple of studies comparing two water vapor instruments on the same airplane with varying levels of agreement. Exemplarily we added the work of Jensen et al (2017) and Kiemle et al. (2008).To our knowledge, the MACPEX intercomparison and this work are the only ones combining five different hygrometers with different measurement techniques on aircraft which provides a unique data set to evaluate the instruments performance and reliability.*

Section 4.3: This section describes the selection of the reference value, and mentions the fact that no single instrument covers the entire range of values observed. This seems to imply that it would be common for some combination or combinations of instruments to be used on this and other German aircraft during other campaigns. Is that the case? If so, which instruments typically fly together? And how to they generally compare in the ranges where they have overlapping measurements?

*Usually there is no fixed combination of hygrometers which fly together on the same plane since payloads are designed specifically for each campaign. Since SHARC is part of the basic instrumentation of HALO it provides $H_2O$ data for most recent campaigns. The comparison to the "frequent flyer" FISH on HALO is typically in the same range as found in this study. Also, AIMS and WARAN are integrated in the same instrument rack, however, their overlap range is very narrow so there is not too much to learn from a detailed comparison.*

Section 6: On page 15, line 20, drift is discussed, but the statement is made that observed relative changes between measurements made by the AIMS and SHARC instruments are not due to drifts in either instrument. As this seems to be difficult to reconcile with the observations, what do the authors suggest is the cause or explanation?

*To check for possible (relative) drifts in one or both instruments, we also looked in the clear sky measurements of both instruments. The question was, whether the observed trend in in-cloud RHi measurements is still existent when looking at the entire data set including clear sky values of RHi and water vapor mixing ratio (to exclude influences from the temperature measurements). That was not the case. What could be a possible explanation is a drift of the instruments relative to each other within certain flights, e.g. due to temperature variations in the aircraft cabin. In that case, the difference of the measurements would depend on the point in time when the (majority of the) clouds was sampled.*

**Minor word changes, etc.:**

Page 1 Line 16 – suggest replacing "turned out to be" with "is" Line 24 – suggest replacing "total mean values even agree" with "and total mean values agree" Line 31 – suggest replacing "deficit" with "error"

*changed*

Page 2 Line 9 – suggest removing "their" Line 32 – suggest adding "but as-yet undocumented" before "campaigns" Line 33 – suggest adding "during AquaVIT-1" before "was improved" and replacing "compared" with "relative"

*changed*

Page 3 Line 15 – suggest replacing "major" with "primary" here and elsewhere.

*changed*

Page 7 Line 25 – "less" should be replaced by "fewer"

*changed*

Page 10 Line 5 – suggest replacing "way" with "well"

*changed*

Page 11 Line 16 – typographical error: "ofmeteorological" should be "of meteorological"

*changed*

Page 13 Line 14 – should "interpolated" be "averaged" ?

*Yes, we changed it.*

Page 14 Line 23 – add comma after "hygrometer"

*changed*

Page 15 Line 28 – replace "IQR" with "interquartile range"

*changed*

Page 16 Line 12 – replace "access" with "assess"

*changed*